# Solid Earth change and the evolution of the Antarctic Ice Sheet

Pippa L. Whitehouse [1], Natalya Gomez[2], Matt A. King [3] &
Douglas A. Wiens[4]

Recent studies suggest that Antarctica has the potential to contribute up to ~15 m of sea-level rise over the next few centuries. The evolution of the Antarctic Ice Sheet is driven by a combination of climate forcing and non-climatic feedbacks. In this review we focus on feedbacks between the Antarctic Ice Sheet and the solid Earth, and the role of these feedbacks in shaping the response of the ice sheet to past and future climate changes. The growth and decay of the Antarctic Ice Sheet reshapes the solid Earth via isostasy and erosion. In turn, the shape of the bed exerts a fundamental control on ice dynamics as well as the position of the grounding line—the location where ice starts to float. A complicating issue is the fact that Antarctica is situated on a region of the Earth that displays large spatial variations in rheological properties. These properties affect the timescale and strength of feedbacks between ice-sheet change and solid Earth deformation, and hence must be accounted for when considering the future evolution of the ice sheet.

The solid Earth, along with the oceans and the atmosphere, exerts a strong influence on the dynamics of the Antarctic Ice Sheet (AIS). The pre-glacial topography of the Antarctic continent determined the location and style of glacial inception ~34 Ma ago[1], whereas today the shape of the bed and the properties of the ice-bed interface exert a first-order control on the contemporary pattern of ice flow[2]. In the intervening period the bed of the AIS has been continuously reshaped by erosion and sedimentation, periodically flexed by glacial isostasy, and abruptly altered by tectonic and volcanic activity, with the latter two processes also playing a role in determining the thermal conditions at the bed (Fig. 1). Basal heat flux affects ice flow via its influence on subglacial hydrology and ice rheology[3], but its spatial variation is currently poorly quantified[4]. Parts of Antarctica are underlain by active volcanoes, notably West Antarctica (Fig. 1). This is of interest because in other regions, e.g., Iceland[5], volcanism has been shown to increase during periods of deglaciation. However, little is known of this phenomenon in Antarctica[6] and a detailed review of this effect is therefore not possible. We instead focus on those processes that are better known and that control long-wavelength changes to the shape of the bed beneath Antarctica – glacial isostasy and erosion – and feedbacks between these processes and ice sheet evolution.

Interactions between ice sheets and the solid Earth have long been studied within the field of Glacial Isostatic Adjustment (GIA)—defined here as the response of the solid Earth and the

---

[1] Department of Geography, Durham University, Durham, UK DH1 3LE. [2] Department of Earth and Planetary Sciences, McGill University, Montreal H3A 0E8, Canada. [3] School of Technology, Environments and Design, University of Tasmania, Hobart, TAS 7001, Australia. [4] Department of Earth and Planetary Sciences, Washington University, St Louis, MO 63130, USA. Correspondence and requests for materials should be addressed to P.L.W. (email: pippa.whitehouse@durham.ac.uk)

global gravity field to changes in the distribution of ice and water on Earth's surface. The first numerical models of GIA were developed in the 1970s[7] but they have received renewed attention over the last decade, reflecting the important role they play in the interpretation of satellite measurements of contemporary ice-sheet change[8]. A number of recent studies have also sought to understand the strength of feedbacks acting in the opposite direction, that is, the impact of solid Earth deformation on ice dynamics and the potential for this deformation to delay or prevent unstable ice-sheet retreat[9,10]. Such feedbacks were first considered in the 1980s[11] but have only recently begun to be implemented into fully-coupled models[12], where the evolving shape of the solid Earth and the depth of the ocean adjacent to an ice sheet grounded below sea level act as fundamental boundary conditions on the dynamics of the ice sheet.

The strength of any feedbacks between glacial isostasy and ice dynamics depends on the rate at which the solid Earth responds to ice-sheet change, which, in turn, depends on the rheological properties of the mantle. Seismic evidence[13] indicates that there are significant spatial variations in mantle properties beneath Antarctica, which suggests mantle viscosities, and hence relaxation timescales, may vary by up to several orders of magnitude from the global mean. Indeed, in the northern Antarctic Peninsula, geodetic evidence[14] suggests that contemporary ice loss is triggering a viscous response orders of magnitude more rapidly than was previously assumed possible in Antarctica – on decadal rather than millennial timescales. Feedbacks on ice dynamics are likely to be enhanced in such regions[15], and the ongoing GIA signal is likely to be dominated by recent ice-sheet change (few millennia or less), as opposed to the ice loss that followed the Last Glacial Maximum (LGM)[16]. Here, we define the GIA signal to be the ongoing response of the solid Earth, the gravity field, and/or relative sea level to past ice-sheet change.

The growth and decay of the AIS is driven by a combination of climate forcing and non-climatic feedbacks, but modelling studies that seek to understand the controls on AIS change often neglect to consider how the ice sheet alters its own boundary conditions. Over the timescale of the last deglaciation, GIA model output[17,18] suggests that the response to surface load change can alter bed slopes across West Antarctica by 0.25–0.4 m/km (values may be an underestimate due to assumptions of strong mantle rheology) and that water-depth change around the margin of the ice sheet can deviate from eustatic by >100 m. Coupled ice sheet-sea level models can now be used to quantify the impact of such changes on ice sheet evolution[15,19]. Over timescales of millions of years, changes to the shape of the ice sheet bed will additionally reflect processes associated with erosion and deposition, the isostatic response to sediment redistribution, and mantle convection. Feedbacks between ice sheet evolution and long-term landscape evolution have been hypothesised[20], but they have not been modelled within a coupled framework. The impacts of long-wavelength isostasy-driven changes to ice sheet boundary conditions are reviewed here, but smaller-scale subglacial controls, such as the material properties of the bed and variations in subglacial hydrology and geomorphology, are discussed elsewhere[2].

In this review we proceed by briefly summarising the current state of knowledge on ice-sheet change and glacial isostasy across Antarctica, before discussing the impact of spatial variations in Earth rheology on glacial isostasy and the impact of GIA on ice dynamics. Feedbacks between ice sheet and landscape evolution are discussed, and we conclude by identifying a number of future research priorities that link ice history, Earth rheology, and the overarching issue of global sea-level change.

## Antarctic ice-sheet change and glacial isostasy

Forward model predictions of GIA-related solid Earth deformation, gravity-field change, and polar motion rely on: a reconstruction of the spatial and temporal pattern of past ice loading, a viscoelastic Earth model that describes the time-dependent response of the solid Earth to surface loading changes, and the iterative consideration of physical feedbacks associated with polar motion and coastline position as they are altered by the deformation of the solid Earth and the gravity field[21,22]. While the third component is well defined by physical theory, Antarctic ice-sheet reconstructions and solid Earth rheology are subject to substantial uncertainty and debate.

GIA modelling can be used to infer past large-scale ice-sheet change via comparison of model output with a range of constraint data relating to sea-level change and solid Earth deformation[18,23]. On a global scale, such data-model comparisons have been used to estimate the timing of ice-sheet growth and retreat over glacial cycles[24,25], but the combined dependence of model output on Earth rheology and ice history leads to non-uniqueness[26], and it remains challenging to determine how ice was partitioned between the different ice sheets[27].

Reconstructions of the LGM AIS suggest an ice sheet larger than present, equivalent to ~5–22 m global mean sea level[28]. This range reflects differences in methodology and the interpretation of data constraints. The most common reconstruction approaches consider either numerical models of ice-sheet dynamics[19,29] or local geological, geomorphological, and geodetic data constraints on past ice extent[30], or a hybrid[31,32]. Data constraints are sparse as they rely on sampling of spatially limited bedrock exposures[33,34] or interpretation of ice core records[35], while numerical ice-sheet models are restricted by a limited representation of reality. Continent-wide ice-sheet reconstructions can be produced by simulating the response of dynamic models to past climate changes[19,31,36–38], but at present there is not enough data to tightly constrain such models. Consequently, there is significant variation in the predicted magnitude and spatial pattern of isostatic deformation across Antarctica due to past ice-sheet change[39] (Fig. 2).

It is important to note that some of the differences in Fig. 2 reflect the timescale of ice-sheet change considered to contribute to the deformation signal; traditional forward models of GIA do not consider ice-sheet change during the last millennium (Fig. 2a-f), but this is accounted for in solutions derived via coupled modelling (Fig. 2g, h) or data inversion (Fig. 2i-l). Altogether, four approaches to determining the present-day GIA signal are represented in Fig. 2, reflecting a number of recent methodological advances. We do not seek to quantitatively assess the accuracy of each GIA solution, partly due to the difficulty of defining validation data sets, but we briefly note the advantages and limitations of each approach in Table 1.

The effect of GIA on geodetic and gravimetric measurements, and the resulting impact on estimates of ice-mass change, has driven much of the recent interest in Antarctic GIA. Contemporary ice-sheet change can be determined from analysis of Earth's time-varying gravity field, as measured by the Gravity Recovery and Climate Experiment (GRACE) satellites between 2002 and 2017[40] and GRACE Follow-On (from May 2018). However, the time-varying gravity field also contains a gravity-change signal associated with past ice-sheet change; GIA modelling is used to determine the 'GIA correction' that should be applied to remove the signal associated with past change[8]. Uncertainties associated with this process remain the dominant source of error in gravimetry-based estimates of the Antarctic contribution to contemporary sea-level rise[41]. However, newer GIA model predictions[17,30,32] yield improved inter-model agreement, as do inverse GIA solutions derived by combining ice elevation and GRACE data[42–44]. Recent estimates for the

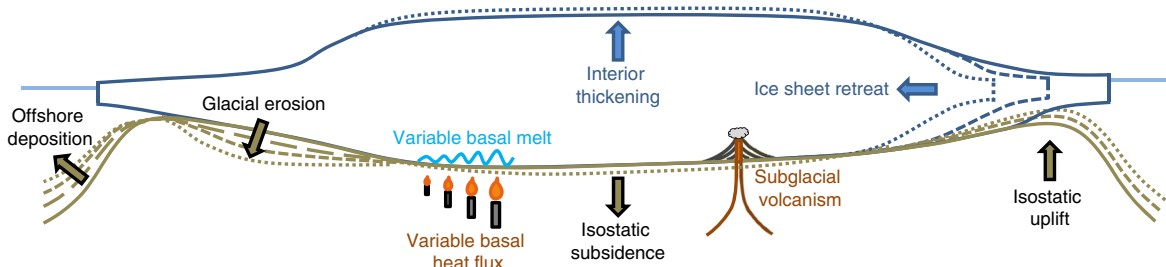

**Fig. 1** Summary of interactions between the solid Earth and the Antarctic Ice Sheet. Local isostatic uplift occurs in response to ice-sheet thinning or retreat, isostatic subsidence occurs in response to ice-sheet thickening or advance. Subglacial volcanism and basal heat flux alter thermal conditions at the base of the ice sheet. Erosion and deposition also trigger an isostatic response (not shown). Increasing time indicated by finer dashed lines

**Fig. 2** Predictions of present-day GIA-related uplift rates across Antarctica derived from forward modelling and data inversion. **a–d** Results derived using GIA forward models that adopt a 1-D Earth model[17,18,30,32,141,145]; in **b** an Earth model that reflects low viscosity West Antarctic mantle rheology is used. **e**, **f** Results derived using GIA forward models that adopt a 3-D Earth model[55,56]; (**e**) uses the same ice model as **c**; (**f**) uses the same ice model as **a**. **g**, **h** Results derived using coupled ice sheet–sea level models that include ice-sheet change through to present[10,87]. **i–l** Results derived from the inversion of geodetic data[42–44,146]; note that **i–l** are reduced in their precision away from data constraints. See original publications for further details

**Table 1 Advantages and limitations of four approaches used to determine the present-day GIA signal across Antarctica**

| Method | Advantages | Limitations |
|---|---|---|
| GIA forward model, radially varying Earth rheology (e.g., Fig. 2a–d) | Can be tuned to fit observational constraints<br>Computationally efficient | Observational constraints are sparse in time and space<br>Cannot account for lateral variations in Earth rheology<br>Do not always reflect ice sheet physics |
| GIA forward model, 3-D Earth rheology (e.g., Fig. 2e, f) | Accounts for lateral variations in Earth rheology | Parameterisation of Earth rheology poorly understood<br>Ice history used within such models not yet tuned to account for influence of lateral variations in Earth rheology<br>Computationally expensive |
| Coupled GIA-ice sheet model with 1-D or 3-D Earth rheology (e.g., Fig. 2g, h) | Accounts for feedbacks between ice dynamics, solid Earth deformation, and sea-level change | Model output not yet tuned to fit observational constraints<br>Lateral variations in Earth rheology not considered in 1-D case<br>Computationally expensive |
| Data inversion (e.g., Fig. 2i–l) | Do not depend on assumptions of ice history or Earth rheology<br>Uncertainty more easily quantified | A range of corrections must be applied to the contributing satellite data sets<br>Some dependence on spatial distribution of data (e.g., GPS sites)<br>Cannot be used to model GIA at other times |

magnitude of the net Antarctic GIA signal vary over the range ~40–80 Gt/yr[45], but basin-level differences remain substantial and in some cases different studies do not even agree on the sign of the mass change in each basin[43], which hampers the advance of glaciological insights into the processes governing present change.

Differences in estimates of the present-day GIA signal are particularly acute in the region of the Amundsen Sea embayment, where forward models have substantially less GIA signal than inverse solutions. This may be due to the fact that the GIA signal in this low mantle viscosity region predominantly reflects significant decadal-to-centennial ice load change that is not accounted for in most forward models[46,47]. For the same reasons, forward models underestimate the signal in the northern Antarctic Peninsula, but then so do inverse solutions because the GIA signal in this region has a shorter spatial wavelength than inverse solutions can resolve (~300 km). Comparison of the new generation of GIA models with GPS-derived uplift rates—which must be corrected for elastic deformation associated with contemporaneous ice-mass change[48,49]—demonstrates that important differences remain, notably in West Antarctica and especially in the Amundsen Sea and northern Antarctic Peninsula regions[39].

## Earth structure and rheology beneath Antarctica

The pattern and magnitude of the solid Earth response to ice-sheet growth and decay is strongly dependent on the rheology of the interior of the solid Earth, and the choice of rheological model has a large effect on the modelled GIA response (see Box 1). For example, ice history is often tuned in tandem with Earth rheology, so adoption of a different rheological model can lead to large differences in the assumed ice history. Earth rheology beneath Antarctica is spatially variable and, as detailed in the next section, this has implications for the behaviour of the ice sheet.

A number of rheological models have been adopted to explain the solid Earth response to surface load change across Antarctica, including linear Maxwell viscoelastic rheology, power law rheology, and Burgers rheology (see Box 1). Regardless of the choice of rheological model, the need for future modelling efforts to consider spatial variations in Earth structure beneath Antarctica is now clear. Seismic studies of the upper mantle beneath Antarctica suggest large lateral variations in material properties[13,50–53], with greater heterogeneity than observed in areas of Northern Hemisphere continental glaciation[54]. East Antarctica shows high upper mantle seismic velocities characteristic of cold cratonic regions worldwide, whereas West Antarctica shows upper mantle

structure consistent with much warmer tectonically active zones (Fig. 3a–c). The range of seismic velocities observed beneath Antarctica provides strong evidence for lateral variations in mantle viscosity, but absolute values are currently poorly known. Preliminary GIA studies that explore the effect of including lateral variations in mantle viscosity reveal significant differences in predicted patterns and rates of deformation across Antarctica[55–58], motivating the need for better constraints on absolute mantle viscosity in this region.

Upper mantle viscosity variations of several orders of magnitude beneath West Antarctica have been quantified by modelling observed GPS uplift in specific settings. Nield et al.[14] identified rapid viscoelastic deformation occurring in the northern Antarctic Peninsula due to a well-observed change in glacial loading resulting from the 2002 breakup of the Larsen B Ice Shelf. Comparison of GPS time series and modelled uplift in the region suggests upper mantle viscosities of between $6 \times 10^{17}$ and $2 \times 10^{18}$ Pa s in the northern Antarctic Peninsula. Similarly, Zhao et al.[59] used the changes in ice load resulting from thinning of Fleming Glacier over recent decades to estimate upper mantle viscosity, 500 km further south along the southern Antarctic Peninsula, to be at least $2 \times 10^{19}$ Pa s. Other estimates of upper mantle viscosity in West Antarctica include $1–3 \times 10^{20}$ Pa s beneath the southwestern Weddell Sea[60] and $4 \times 10^{18}$ Pa s for the Amundsen Sea Coast[47]. These represent variations of two orders of magnitude. In contrast, estimates of spatially-averaged upper mantle viscosity beneath the whole of Antarctica range from $2 \times 10^{20}$ to $1 \times 10^{21}$ Pa s (ref. [16,17]), which are similar to estimates of upper mantle viscosity in Fennoscandia[61]. The emerging picture suggests cratonic East Antarctica is characterised by higher upper mantle viscosity than West Antarctica, with exceptionally low upper mantle viscosity, on the order of $10^{18}$ to $10^{19}$ Pa s, beneath some regions of West Antarctica.

Existing constraints on mantle viscosity across Antarctica draw on our ability to measure the solid Earth response to known surface-load change. In regions where this is not possible, the three-dimensional shear velocity structure of the upper mantle beneath Antarctica can be used to estimate the lateral and depth variation of viscosity. Although there is no direct physical correspondence between shear velocity and viscosity, their variation in the upper mantle is largely controlled by temperature[62,63]. Ivins and Sammis[64] formulated an approach for converting shear velocity anomalies to viscosity anomalies relative to a global reference 1-D viscosity model, using the observed scaling of shear velocity and density anomalies to infer the temperature scaling, and then using olivine diffusion creep (linear) flow laws to

---

**Box 1 | Rheological models of the solid Earth**

The response of the Earth to changing loads has generally been described using a linear Maxwell viscoelastic rheology, with an instantaneous elastic response superposed on a longer-term Newtonian viscous relaxation[149,150]. The majority of GIA models, including most coupled ice sheet–sea level models, adopt this simple rheology and consider a spherical Earth, with an elastic lithosphere, a layered viscoelastic mantle, and an inviscid core. Crucially, such models typically include no lateral variation in rheological structure. However, differences in the response of the Earth to surface loading around the world suggest regional variations in rheological properties. Laboratory experiments on mantle materials, primarily olivine, show that the mantle can respond to long-term loading with either diffusion creep, corresponding to linear viscosity, or dislocation creep, corresponding to a power-law viscosity[62,151,152]. Although both mechanisms operate simultaneously in the mantle, deformation will be controlled by the weaker mechanism at any given location, with higher stress and larger grain size favouring dislocation creep, i.e., a non-linear response[153]. It is commonly thought that dislocation creep dominates at shallow depth in the upper mantle, as indicated by xenoliths[154] and significant seismic anisotropy[155], transitioning to diffusion creep at depths greater than 200–300 km (ref. [152]). It is difficult to clearly delineate the regions of the mantle dominated by linear and power-law viscosities because the rheologies depend on stress and the poorly constrained parameters of water content, grain size and activation volume[153]. Power-law rheologies can be introduced into GIA models using a composite rheology, where low stress portions of the model adopt a linear Maxwell rheology and high stress portions adopt a power law rheology assuming some transition stress[156]. Alternatively, strain is assumed to be the sum of diffusion and dislocation creep, as calculated using laboratory-based flow laws assuming parameters such as grain size and temperature[56,153,157].

Recent studies show that the transient relaxation following major earthquakes is generally best fit by a Burgers (biviscous) rheology with two characteristic relaxation times or effective viscosities[158,159]. These observations have resulted in increased interest in the use of Burgers rheology in GIA studies[27,160]. However, similar to the power-law case, there are few data to constrain these more complex models and most recent GIA models continue to use a Maxwell rheology[17,30,32].

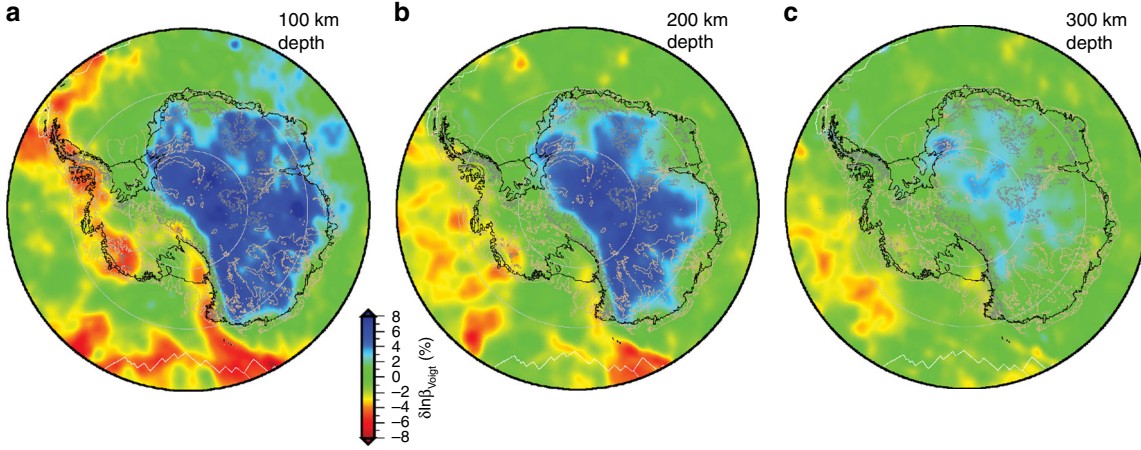

**Fig. 3** Spatial variations in Earth properties in the upper mantle beneath Antarctica. Seismic shear wave velocity perturbations at depths of (**a**) 100, (**b**) 200 and (**c**) 300 km beneath Antarctica derived from an adjoint inversion of teleseismic waveforms recorded by seismic stations south of 45°S (ref. [52]). Colours indicate observed Voigt average velocity anomalies relative to global reference model STW105 (ref. [147]). Regions of negative anomalies (slower than global average velocities) suggest higher temperatures and hence likely lower viscosities

constrain the viscosity perturbation. Wu et al.[65] proposed a similar method that uses experimentally determined temperature derivatives of shear velocity, including the effect of anelasticity. An alternative method is to estimate the temperature structure of the mantle from the shear velocity structure, and then use a composite rheology that computes the strain associated with both diffusion and dislocation creep mechanisms[56]. Since the dislocation creep rheology is non-linear, the calculated effective viscosity of the mantle will depend on the stress field. One key assumption in any of these methods is that the observed seismic velocity anomalies result entirely from temperature anomalies, whereas it is well known that a portion of the velocity perturbations result from (spatially variable) compositional anomalies[66]. Wu et al.[65] adjust their viscosity structure by the percentage of the velocity anomalies thought to be caused by thermal variations, which in their study was estimated to be between 65% and 100%. An alternative approach would involve correcting for (poorly known) spatial variations in mantle composition.

Using the approach outlined in Wu et al.[65], the estimated mantle viscosity beneath Antarctica, at depths of 100 and 250 km, varies by 2–3 orders of magnitude across the continent (Fig. 4),

with extremely low viscosity predicted beneath the Antarctic Peninsula, the Amundsen Sea coast, Marie Byrd Land, and the Transantarctic Mountain Front. The implications of these viscosity variations, and in particular the anomalously low viscosities beneath parts of West Antarctica, are explored in the next section.

## Feedbacks between GIA processes and ice dynamics

It has long been recognised that the evolution of ice sheets is influenced by the geometry and deformation of the underlying solid Earth[67], and that the stability and dynamics of marine ice sheets (ice sheets which rest on ground below sea level) are sensitive to the depth of water at their grounding lines, i.e., the point where they begin to float[68–70]. Marine-terminating ice sheets such as Antarctica lose most of their mass following the flow of grounded ice across their grounding lines into floating ice shelves (Fig. 5). The ice flux across the grounding line is very sensitive to the thickness of ice there, and the thickness is in turn proportional to the depth of water such that a small increase in water depth at the grounding line leads to a large reduction in grounded ice[70]. Marine ice sheets are widely thought to be prone to runaway retreat when resting upon beds that slope down towards the interior of the ice sheet, i.e., reverse bed slopes[70], as is

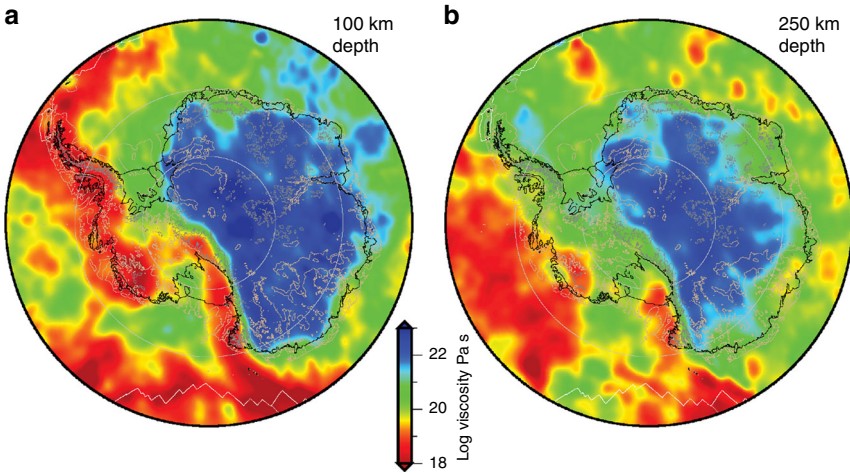

**Fig. 4** Spatial variations in estimated upper mantle viscosity beneath Antarctica at depths of (**a**) 100 km and (**b**) 250 km. Viscosity is estimated using the seismic shear velocity model presented in Fig. 3 and the method outlined in Wu et al.[65], as discussed in the text. These results are derived using a dry diffusion creep rheology[62] and the IJ05-R2 (ref. [30]) reference viscosity model, assuming that the seismic velocity variations are entirely due to temperature, except that a correction for depleted cratonic mantle is applied to the East Antarctic lithosphere

the case for West Antarctica and much of East Antarctica. This canonical argument is based on the idea that when the grounding line retreats into deeper water, the ice loss at the grounding line increases, leading to further retreat (Fig. 5b). However, this reasoning does not account for the fact that as ice mass loss occurs, the solid Earth rebounds, and the gravitational attraction of the ice on the surrounding water is diminished, lowering the local sea surface (Fig. 5c). The bathymetry shallows and bed slopes are altered near the margins of the ice sheet where ice mass loss is occurring. Within a modelling framework this local shallowing of water reduces the loss of ice across the grounding line, acting to stabilise the modelled ice sheet[71], slowing, and in some cases halting, migration of the modelled grounding line along reverse bed slopes[12]. Ongoing viscous uplift of the bed following a halt in ice sheet retreat can also initiate readvance of the grounding line in marine areas (e.g., Fig. 6d, ref. [15]).

Water depth changes can also influence the degree to which ice shelves are able to stabilise the ice sheet. Ice shelves play an important role in the stability and evolution of marine ice sheets by providing resistance to the flow of grounded ice across the grounding line. The spreading of ice shelves is inhibited by friction along their sides and base, particularly where the ice shelf becomes locally grounded on bumps or pinning points in the bathymetry, forming ice rises[72]. A local decrease in water depth can enhance grounding of the ice shelf at ice rises, stabilising the ice sheet, while an increase in water depth can lead to ungrounding at the ice rise, enhancing flow across the grounding line of the ice sheet. Modelling studies of the AIS have shown that accelerated retreat and thinning and eventually large-scale collapse of marine sectors of the ice sheet can occur when the surrounding ice shelves break up[73], but that if ice shelves are able to re-ground, this can have a stabilising effect on the ice sheet[74]. Furthermore, changes in bathymetry may have implications for ocean circulation and heat transfer under the ice shelves. The role of these and other ice shelf processes in controlling the dynamics of the grounded portion of the ice sheet are discussed in more detail in a companion paper by Smith et al. (manuscript submitted).

Initial coupled modelling studies, using simplified flowline ice-sheet modelling and bedrock geometry, demonstrated the stabilising feedback of sea-level changes on marine ice sheets[12,71]. Recently, a series of more realistic coupled models have been developed that capture Antarctic ice sheet and ice shelf dynamics,

global sea level and solid Earth deformation, and the interactions between these systems[9,10,15,19,36]. It has been shown that GIA-related sea-level and solid Earth changes, including changes to the slope of the underlying bed, alter the stress field of the ice sheet in a way that acts to dampen and slow past[19,75] and future[9,10,76] ice-sheet growth and retreat in Antarctica (Fig. 6a, c). An important process that is also accounted for in these coupled models is the feedback between isostatically-driven ice surface elevation change and surface mass balance[75,77,78].

The strength of the feedback between GIA processes and ice dynamics depends on the rheology of the solid Earth. Models representing the palaeo[79–81] and long-term future evolution of ice sheets over millennia[73,82] generally account for ice-load-driven Earth deformation by adopting simplified treatments such as the Elastic Lithosphere Relaxing Asthenosphere (ELRA) model, which treats the asthenosphere as a time-lagged relaxation towards equilibrium[11,73,83], or a model with flow in a viscous half-space below an elastic plate[82,84]. The Earth rheology models adopted in the newly-developed fully-coupled models described above[19,36,75] capture more realistically the full multi-normal-mode response of the Earth to both ice and water loading, thus enabling the computation of gravitationally self-consistent variations of the sea surface, and Earth rotational effects. By solving the sea-level equation[7], these studies also account for migrating shorelines, including migration into regions previously occupied by marine-based ice. Ice model simulations[15] over the last deglaciation incorporating an ELRA bed model (black line) and the full sea-level coupling (blue and red lines) are shown in Fig. 6a. Note that the bed topography at the start of each simulation shown in Fig. 6a will be different; this is to ensure that the final modelled topography is close to the modern observed topography in each case. Due to the complexity and spatial variations of bedrock geometry, ice dynamics and climate-ice interactions, the importance of GIA processes on ice sheet evolution cannot be quantified universally and must be considered on a case-by-case basis for different regions, time periods, and climate forcings.

When considering short-term change, ice-sheet models designed to simulate decadal to centennial-scale transient ice dynamics in response to future warming[85,86] often do not include bedrock deformation and sea-level changes. Simulated Antarctic ice volume changes under moderate future climate warming with fixed bed and sea surface are compared with results from a coupled model[9] in which these surfaces are allowed to vary (Fig. 6c). The impact of sea-level changes on ice dynamics has

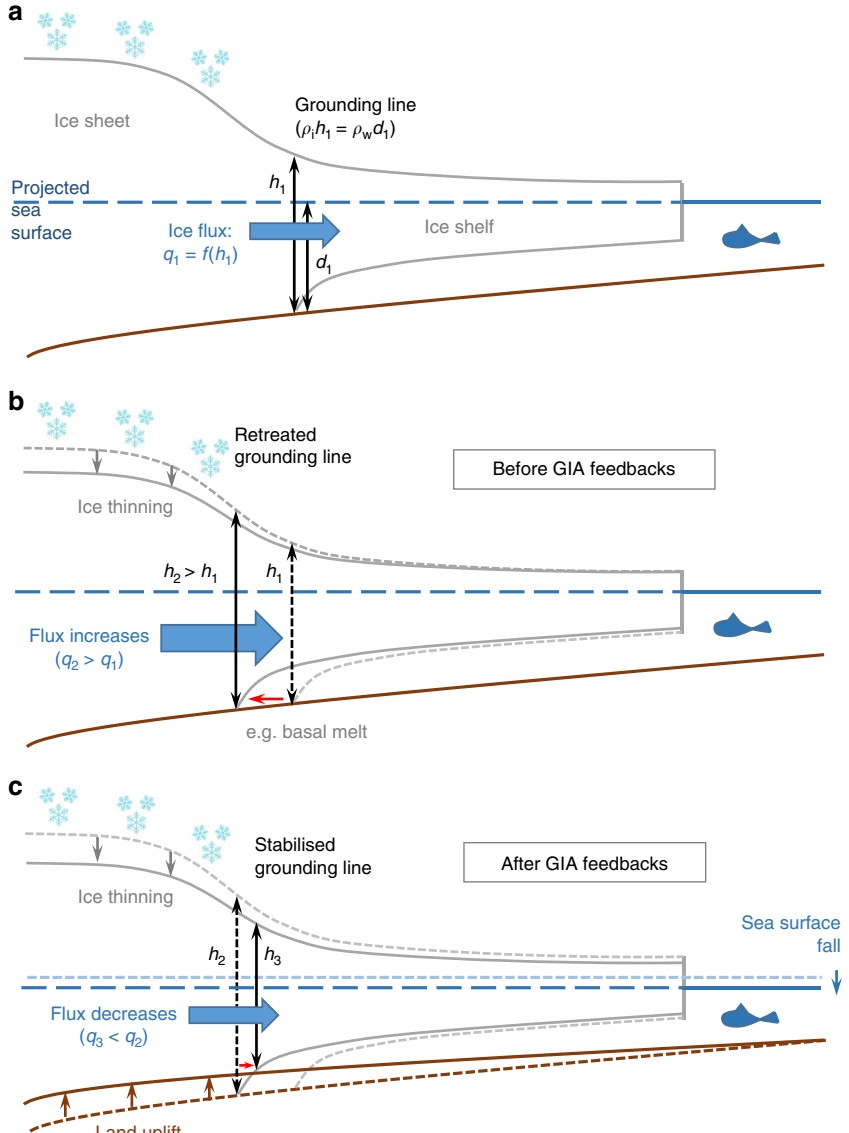

**Fig. 5** Stabilising effect of GIA on ice dynamics. **a** The grounding line is the location where the ice sheet begins to float. Flux across the grounding line ($q_1$) depends on ice thickness at the grounding line ($h_1$). The position of the grounding line also depends on water depth at the grounding line ($d_1$), the density of ice ($\rho_i$), and the density of seawater ($\rho_w$). **b** Before GIA feedbacks: on a reverse bed slope grounding line retreats into deeper water (red arrow) due to, e.g., a reduction in ice shelf buttressing. Increase in ice thickness at the grounding line ($h_2 > h_1$) results in an increase in ice flux across the grounding line ($q_2 > q_1$). If ice input from upstream does not increase then the ice sheet thins, triggering further grounding line retreat. **c** After GIA feedbacks: following grounding line retreat, grounded ice loss triggers land uplift and sea surface lowering. The resulting decrease in water depth and hence ice thickness ($h_3 < h_2$) at the grounding line results in a decrease in ice flux across the grounding line ($q_3 < q_2$), which acts to stabilise the position of the grounding line. Initial configurations shown by dashed lines, new configurations shown by solid lines

previously been considered negligible on these timescales, in particular under strong warming, because the viscous response of the Earth is only considered relevant on millennial timescales and longer. However, as discussed above, the Earth structure underneath the AIS is highly variable, and viscosities may be as low as $10^{18}$ Pa s beneath parts of West Antarctica, leading to substantial (i.e., metres to tens of metres of) viscoelastic uplift occurring on centennial or even decadal timescales[14,47], with consequent implications for ice sheet evolution.

Recent coupled modelling studies have begun to quantify the sensitivity of predicted ice dynamics, GIA and crustal deformation in Antarctica, as well as the AIS's contribution to past and future global sea-level change, to the adopted radial viscosity structure of the solid Earth[9,10,15]. Note that, in a coupled modelling context, altering the Earth structure influences GIA

predictions both directly by altering the timing and geometry of the Earth's response to surface loading, and indirectly by changing the ice loading itself. The blue and red lines in Fig. 6a-d compare coupled model simulations[9,15] over the last 40 ka, during a Pliocene warm period, and into the future under moderate and strong climate warming scenarios (see figure caption for details) adopting two different radially-varying models of Earth viscosity and lithospheric thickness within the coupled model. One is a relatively high viscosity Earth model with a thick lithosphere (HV), similar to models adopted in global GIA studies, and the other is representative of the Earth structure beneath much of West Antarctica, having a thinned lithosphere and a zone of low viscosity down to 200 km in the upper mantle (LVZ). For each of the time periods considered in Fig. 6, the timing and extent of ice-sheet retreat is sensitive to the adopted Earth

rheology, and the predicted contribution of Antarctica to sea-level rise is lowered in the LVZ simulation as compared to the HV simulation. In the LVZ case the solid Earth responds faster to ice loading changes and deformation is localised to near the edges of the ice sheet where the ice loss occurs. The resulting sea-level fall at the grounding line can more effectively act to slow ice-sheet retreat as compared to the simulations with the HV Earth model.

A comparison of the future simulations shown in Fig. 6c, d highlights that the influence of Earth structure on ice sheet evolution depends on both the strength of the climate forcing and the physics adopted in the ice-sheet model. For a moderate climate warming, uplift of the LVZ Earth model preserves much of West Antarctica as compared to the simulation with the HV Earth model (Fig. 6c). While, for the simulation where strong RCP 8.5 climate warming is applied and new rapid-retreat-promoting ice physics are added (hydrofracturing and cliff failure[73]), West Antarctica collapses early on regardless of the choice of Earth model, and differences in the ice sheet evolution between the LVZ and HV simulations occur mostly in East Antarctica (Fig. 6d).

Given the sensitivity of the ice-Earth-sea level system in Antarctica to differences in radially varying Earth structure (Fig. 6), and the known variability in Earth structure beneath Antarctica (Fig. 3), no single, radial Earth model is able to accurately represent all of Antarctica and hence consideration of lateral variations in Earth structure is motivated. Incorporating lateral variations in Earth structure into a coupled ice sheet–sea level model represents a large jump in computational cost. Gomez et al.[87] developed the first coupled ice sheet–sea level model that incorporates 3-D variations in Earth structure and applied it to model Antarctic evolution over the last deglaciation. They show that substantial localised differences can arise in ice cover, sea level and crustal movement, which introduces substantial uncertainty into the GIA corrections that should be applied to contemporary geodetic observations.

## Feedbacks between ice-sheet change and landscape evolution

The previous section describes the impact of ice-load-driven changes to the elevation of the solid Earth surface and gravity field on ice dynamics. Many ice-sheet modelling studies account for ice-driven isostatic adjustment[88], but over multiple glacial cycles several other processes also alter the underlying topography. Thermal subsidence following tectonic extension has lowered central West Antarctica by several hundred metres over the last ~34 Ma (ref. [89]) and changes in dynamic topography over the last 3 Ma are postulated to have altered the stability of some sectors of East Antarctica[90]. However, since the inception of the AIS, glacially-driven erosion and sedimentation, and the accompanying isostatic response[91–94], have been the main drivers of topographic change across Antarctica. This topographic change has, in turn, played a role in controlling the sensitivity of the ice sheet to climate forcing. This section focuses on the extent to which feedbacks between the ice sheet and the solid Earth have shaped the long-term evolution of both.

Prior to 34 Ma, Antarctica was characterised by a fluvial landscape[95] and higher mean topography[96]. Dated offshore sediment packages are testament to the volume of material that has been removed from Antarctica since 34 Ma (ref. [97]). Back-stripping techniques can be used to unpick the history of progradation and isostatic subsidence, and hence reconstruct the palaeotopography of the continental shelf[98]. It is more challenging to reconstruct onshore palaeotopography due to the difficulty of determining the volume of material that has been removed, and the source of that material. By using assumed drainage pathways[89] or process-based erosion modelling tuned to match offshore sediment volumes[92], and accounting for the isostatic response to ice and ocean loading, as well as sediment erosion and deposition, bounds can be placed on the pre-glaciation topography of Antarctica[96,99]. Current palaeotopography reconstructions have been used to suggest that the West Antarctic Ice Sheet could have formed much earlier than previously thought, during the Eocene-Oligocene transition[100]. However, when two palaeotopography end members are used to define boundary conditions for ice-sheet growth under 'cold' mid-Miocene conditions, the difference in modelled ice volume is equivalent to 20 m sea level[101], with the shallower topography leading to the growth of a larger ice sheet. It is clear that the configuration of the ice sheet is highly sensitive to the underlying topography, but the question remains as to how Antarctic topography has evolved, and the degree to which this evolution was coupled to ice-sheet change via glacial erosion and the resulting isostatic response.

To understand the role of internal, i.e., non-climatic, processes in controlling long-term ice-sheet change it is necessary to consider feedbacks between ice dynamics, erosion, deposition, isostasy, and water depth change. Extending the coupled ice sheet - sea level modelling approach described above to account for landscape evolution processes would allow the testing of hypotheses that seek to explain the dramatic changes that took place as Antarctica evolved from a terrestrial to a marine ice sheet against a background of long-term cooling. Specifically, between 34 and 14 Ma the volume of the AIS fluctuated significantly[102], then at 14 Ma there was a switch to cold, polar conditions[103], which resulted in the establishment of a cold-based, less erosive ice sheet[104]. A range of hypotheses have been proposed to explain this transition to the current 'icehouse' world[105,106], but it remains to be tested whether landscape evolution played a role, either through a rapid change in topographic boundary conditions or by causing the system to pass some internal threshold as large portions of the ice sheet became marine grounded.

Ice sheet-driven processes that will have had an impact on ice-sheet dynamics include progradation of the continental shelf and long wavelength erosion and deposition[2]. The resulting isostatic response of onshore uplift and offshore subsidence will have focused erosion on the inner shelf[92], leading to the development of a reverse bed slope that is too deep to support ice-sheet advance[98,107] and is susceptible to unstable grounding line retreat[68]. These processes are conceptually simple, but they will have been subject to control by spatially variable sea-level change. As the mean topography of Antarctica decreased through erosion, the ice sheet will have become more sensitive to sea-level change[108] and ocean forcing[109]. Applying appropriate boundary conditions to understand long-term ice-sheet change is challenging due to uncertainties associated with palaeotopography and the growth of the Northern Hemisphere ice sheets, but initial studies that have explored the impact of spatially variable sea-level change on the pre-Pleistocene ice sheet highlight the importance of accounting for peripheral bulge growth when considering ice[110] and ocean[111] dynamics, and the damping effects of ice sheet - sea level feedbacks in regions of weak mantle viscosity[15]. The impact of spatial variability in Earth rheology on coupled ice sheet - landscape evolution has not yet been investigated.

Regions of the AIS that are currently located on over-deepened reverse bed slopes have been suggested to be fundamentally unstable[86]. The details of how the ice sheet and the underlying topography reached this state are not known, but it is clear that feedbacks will have played a role as the fluvially- and tectonically-controlled topography of 34 Ma was eroded and warped into its current configuration[94], and it has been suggested that future landscape evolution will render the ice sheet even more

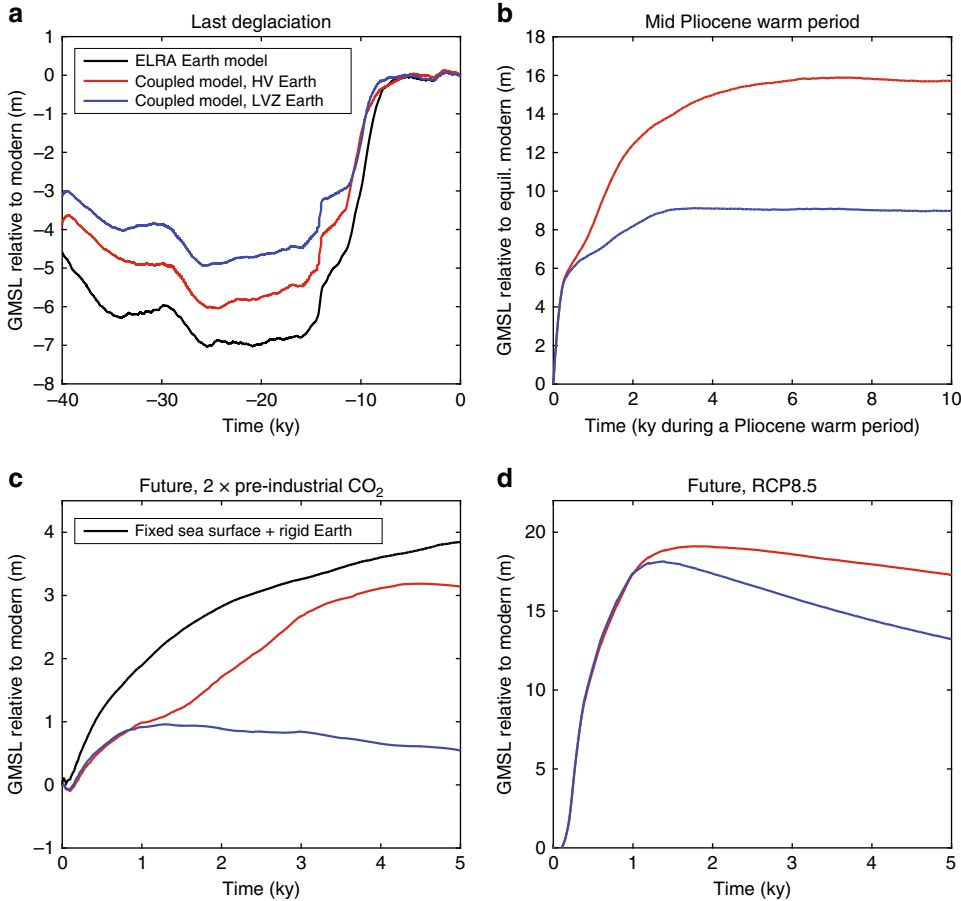

**Fig. 6** Impact of sea-level changes and solid Earth deformation on Antarctic past and future ice sheet evolution. Frames show model-predicted contribution from the AIS to global mean sea level (GMSL) (**a**) over the last 40 ka relative to the modern, (**b**) during a 10 ka mid Pliocene warming of 2 °C relative to an initial state similar to the modern except that the Earth system has fully equilibrated from the last deglaciation, and (**c**, **d**) under future climate warming relative to a modern pre-industrial state at 0 model years in which (**c**) $CO_2$ is doubled and the oceans are warmed by 2 °C at the start of the simulation, and (**d**) RCP 8.5 climate warming is applied and additional mechanisms promoting ice loss are included. The black line in **a** shows the results of a simulation that adopts an ELRA bedrock deformation model (relaxation time 3 ka) and no sea-level coupling. The black line in **c** shows the results of a simulation in which the positions of the solid Earth surface and sea surface are fixed. Red and blue lines show output from a coupled ice sheet–sea level model that includes the feedback of solid Earth deformation and sea surface height changes on ice dynamics. These simulations adopt two different radially varying Earth models: "HV" (red lines) has a lithospheric thickness of 120 km and upper and lower mantle viscosities of 0.5 and $5 \times 10^{21}$ Pa s, respectively; "LVZ" (blue lines) has a 50 km thick lithosphere, a low viscosity zone of $10^{19}$ Pa s down to 200 km depth, and upper and lower mantle viscosities below that of 0.2 and $3 \times 10^{21}$ Pa s, respectively. Frames (**a**), (**b**) and (**d**) were created from results of Pollard et al.[15], and frame (**c**) was created from the results of Gomez et al.[9]

unstable[20]. Models that seek to explore the past and future long-term evolution of the AIS should account for evolution of the underlying topography due to both sediment redistribution and isostasy, ideally within a coupled framework. The development of such models will improve understanding of sediment transport pathways, permitting stronger conclusions to be drawn from provenance studies[112]. Better quantification of bedrock erosion rates and offshore sediment packages are important targets for improving our understanding of feedbacks between ice sheet and landscape change.

## Outstanding problems and future outlook
The previous section highlights a number of advances that are required to further our understanding of the long-term evolution of the AIS and the role of changing topography. These include the development of more sophisticated numerical models that consider feedbacks between ice dynamics, glacial erosion, isostasy, and global sea-level change. Such processes will also play a role in controlling contemporary and future ice dynamics but, as highlighted above, quantifying present change is exacerbated by the

need to interpret geodetic observations in terms of the response to both past and present ice-sheet change. In this section we discuss several areas that should be prioritised as we seek to better quantify the GIA signal across Antarctica and hence understand the processes responsible for contemporary ice-sheet change.

Characterisation of absolute mantle viscosity is preliminary. Beneath West Antarctica low viscosity mantle approaches isostatic equilibrium more quickly compared with global-average timescales. The consequence of this is that the present-day rate of adjustment will depend heavily on recent, and relatively localised, surface loading changes[87]. Characteristic relaxation times of a Maxwell fluid may be approximated by dividing the viscosity by the shear modulus ($\sim7 \times 10^{10}$ Pa for the upper mantle). The relaxation time of mantle material with a viscosity of $10^{19}$ Pa s is therefore only ~5 years, or 1–2 orders of magnitude faster than global averages. Actual relaxation times will be somewhat larger due to the presence of higher-viscosity layers within the rheological profile, but this calculation illustrates that, for regions underlain by low viscosity mantle, detailed knowledge of glacial load changes over the past decades to centuries is needed to quantify the GIA signal in regions of low viscosity[113]. For

viscosities of $10^{18}$ Pa s or lower the viscous relaxation almost occurs contemporaneously with the surface load change and, by implication, the elastic deformation, with the viscous component being around an order of magnitude greater than the elastic component[14,47]. Such a rapid response is at odds with the traditional idea that the present-day GIA signal is related to post-LGM ice loss, although, given the much longer relaxation time of the lower mantle, some component of the present-day GIA signal is likely still associated with large-scale, post-LGM ice-sheet change.

Across East Antarctica, spatial variations in Earth rheology are currently poorly constrained, as they are across all offshore regions, due to significant uncertainty in Earth structure resulting from the absence of seismic stations. Across all Antarctica, uncertainties also exist in relation to the rheological law that should be used to describe mantle deformation. Although traditional GIA models do not parameterise viscosity to be directly related to the physical properties of the mantle, physically-based approximations to mantle creep processes are being implemented in new models[56], and these require quantification of mantle temperature, grain size, and water content, which have varying measurement uncertainties. Mantle temperatures, and hence viscosities, may be inferred from seismic velocity perturbations, but different approaches yield different results. When using a power-law rheology, the stress in the mantle prior to the change in surface load also plays a role in defining the viscosity, but this is often taken as zero[114]. Such stresses are expected to be largely a function of long-term mantle convection processes but could be more complex in the shallow mantle and asthenosphere due to ongoing isostatic relaxation or earthquake-related deformation.

Constraints on ice loading history are extremely sparse across all time periods and locations (Fig. 7). However, this problem is particularly acute for late Holocene loading changes across West Antarctica and post-LGM changes in East Antarctica. Very few data record changes to the AIS from the late Holocene up until the commencement of the satellite record[115], although there is evidence of a dynamic West Antarctic Ice Sheet[116–119] and large changes in net accumulation during this period[120,121]. Progress can be made by studying the internal structure of the ice sheet to determine past flow patterns[122,123], but at present, global and Antarctic-focused GIA models generally assume no ice-load change over the last 1–2 ka (see refs. [14,17,116,124] for some recent regional exceptions). Combined with low mantle viscosities in much of West Antarctica, this means that the predicted upper-mantle component of present-day deformation is likely erroneous. For vast sections of East Antarctica almost no data exist on past ice extent and retreat history[125] (Fig. 7). Furthermore, there are few data to constrain the spatial extent and history of the post-LGM margin due to very limited bathymetric sampling. These limitations provide the motivation for new fieldwork and the reanalysis of historic and high resolution palaeo datasets[59,126].

Data with which to validate or constrain the GIA signal across Antarctica are also sparse (Fig. 7). Less than 20 records of past sea-level change exist across the continent (e.g., through dating raised beach terraces), and these are presently limited to the last 12–15 ka (ref. [31]). It is possible to extend data to earlier time-periods through the study of sediment from submerged offshore basins[127]. Opportunities exist for new absolute gravity[128], InSAR[129] and, especially, GNSS measurements at rock outcrops with large sections of East Antarctica sparsely or un-observed. Continuous measurements allow both increased precision but also, in regions of low mantle viscosity, probing of mantle properties by time-varying surface loading[14,59]. The largely unexploited horizontal deformation field promises new insights into local or regional-scale deformation patterns of both elastic and viscoelastic processes[59,130], although a robust approach to

remove tectonic plate motion and post-seismic deformation prevents continent-wide analyses at present[131,132]. A methodology is not yet available to precisely measure bedrock displacement under the present ice sheet or offshore, and this precludes model validation for these regions.

An important supplement to GNSS measurements of deformation at discrete bedrock outcrops is provided by inverse estimates of the spatially continuous deformation field[42,43]; such estimates are useful in separating competing conventional 'forward model' predictions of the GIA signal[124]. While possessing their own inter-solution variation due largely to differences in altimeter snow-densification corrections, inverse estimates tend to differ more from the spatial patterns of deformation predicted by current forward models (Fig. 2). In regions where mantle viscosities are thought to be ~$10^{18}$ Pa s or lower, viscoelastic deformation in response to contemporary surface load change will perturb the deformation and gravity fields due to the short response time of the mantle, but the effect of this signal on inverse solutions is yet to be addressed.

Quantification of model prediction uncertainty is immature with most attempts limited to sampling a generally small set of Earth models. Ice history uncertainty is rarely taken into account, in part due to the sparse information available from which to sample probabilistically[133] or lack of rigorous measurement uncertainties to propagate formally[134]. This prevents the robust propagation of GIA uncertainties into other quantities that make use of GIA model predictions, for example GRACE-derived ice mass balance estimates.

A practical problem for those wishing to employ viscoelastic models is the absence of open source software that includes state-of-the-art model physics. While open source software are available and widely used for purely elastic[135,136] or viscoelastic[137,138] solutions, the viscoelastic models do not solve the full sea-level equation[139], which requires consideration of gravitationally self-consistent meltwater redistribution on a rotating, spherical Earth with polar-motion feedbacks and migrating coastlines, and they do not have the capacity to account for 3-D Earth structure, compressibility and a selection of rheologies (e.g., transient, linear and power-law). Given that there is just one real Earth, a standardised viscoelastic software framework that allows the consistent treatment of GIA and post-seismic viscoelastic deformation would have distinct advantages; it would pave the way for new insights into Earth's rheology, provide a framework to advance Earth's viscoelastic predictability, and help resolve ongoing debates regarding model robustness[140–142].

## Summary

Recent modelling advances have highlighted the importance of understanding the role of the solid Earth in moderating the response of the global ice sheets to past and future climate change[9,10,19,36]. It has long been understood that the growth and decay of an ice sheet will alter the shape of the solid Earth and result in spatially variable sea-level change[7]. We are beginning to understand how those changes affect the dynamics of ice sheets[71], but further work is needed to quantify feedbacks between ice sheet evolution and landscape evolution before we can fully explain the long-term evolution of the Antarctic Ice Sheet.

Coupled models that consider interactions between ice sheets and the solid Earth via dynamic modelling of both systems represent a new type of model that can provide insight into the extent, timing, and mechanisms of past ice sheet evolution and sea-level change. Such models also provide a new tool for understanding the link between past climate and global sea-level change. Coupled models have been used to demonstrate that the

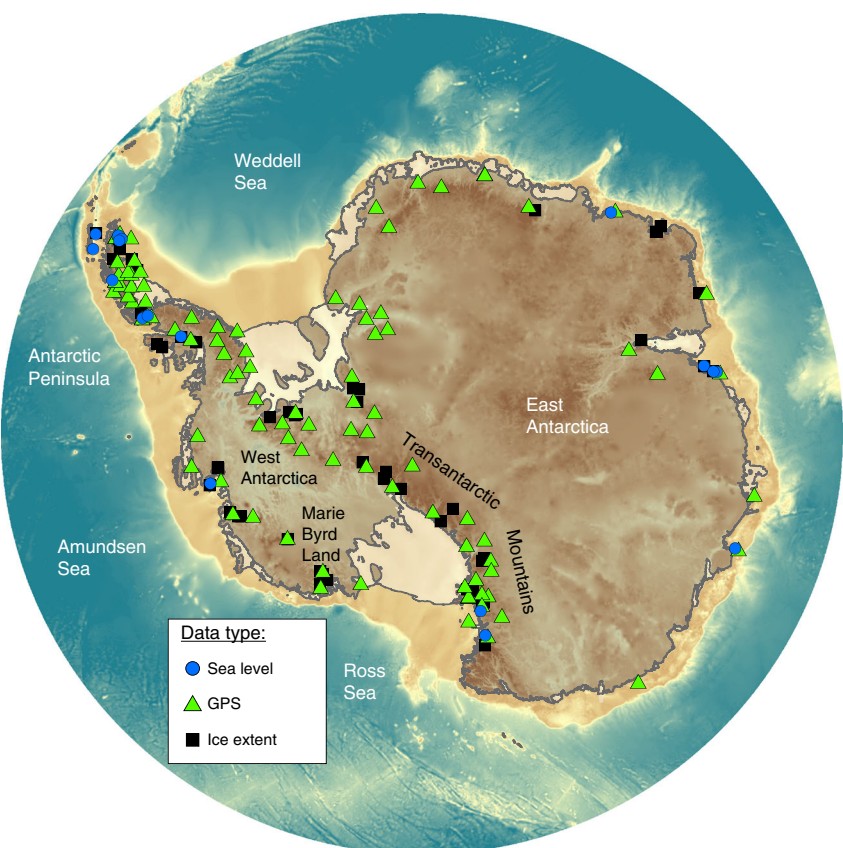

**Fig. 7** Location of GPS sites and data relating to past ice extent and sea level around Antarctica. Information on past ice extent can be derived from surface exposure dating of erratics and bedrock outcrops; locations extracted from the ICE-D Antarctic Database (http://antarctica.ice-d.org/). Sea-level data are documented in Whitehouse et al.[17], GPS data are archived by UNAVCO (http://www.unavco.org/data/gps-gnss/gps-gnss.html). Bathymetry from BEDMAP2[148]

solid Earth response to ice loss, and the accompanying fall in sea level, will have a stabilising effect on grounding line dynamics[71]. This mechanism has the potential to slow or even halt grounding line retreat along a reverse bed slope, depending on the rate of solid Earth rebound. Such negative feedbacks have been modelled in association with ice loss across West Antarctica[9,10], but crucially the rheology of the underlying mantle is poorly constrained, and so the strength of the feedback cannot currently be quantified.

Interdisciplinary approaches to determining mantle rheology reveal large differences between East and West Antarctica[52,56], with low viscosity mantle inferred to lie beneath West Antarctica. The short mantle relaxation time associated with such regions means that estimates of the contemporary GIA signal derived by forward modelling will be biased if they do not account for ice-sheet change over the last few millennia, with implications for estimates of current ice mass balance[8]. Improved quantification of late Holocene ice history, as well as tighter constraints on mantle rheology, are needed to reduce uncertainty on the contemporary GIA signal across Antarctica. Low viscosity mantle will also enhance the strength of the stabilising effect of GIA on grounding line dynamics, highlighting the importance of considering such feedbacks when modelling the future evolution of the Antarctic Ice Sheet. A number of studies have successfully incorporated 3-D Earth structure into GIA models beneath modern ice sheets[56,143,144], and it is now apparent that coupled modelling and inclusion of 3-D Earth structure should both be considered when modelling solid Earth-cryosphere feedbacks[15]. Such models are being developed[87] but further interdisciplinary work, combining modelling and observational approaches, is needed to calibrate such models and better understand controls on the evolution of the Antarctic Ice Sheet.

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

## Acknowledgements

We thank the following people for providing model output shown in Fig. 2: Erik Ivins, Dick Peltier, Lev Tarasov, Geruo A, Wouter van der Wal, Ingo Sasgen, Brian Gunter, and Alba Martín-Español. P.L.W. was supported by a UK Natural Environment Research Council (NERC) Independent Research Fellowship (NE/K009958/1). N.G. was supported by National Sciences and Engineering Research Council (NSERC), the Canada Research Chairs Program and McGill University. M.A.K. was supported by the Australian Research Council Special Research Initiative for Antarctic Gateway Partnership (Project ID SR140300001). This article is a contribution to the SCAR SERCE program.

## Author contributions

P.L.W. drafted the structure of the article, wrote the introductory and summary text, and the section on long-term landscape evolution. N.G. wrote the section on feedbacks between GIA and ice dynamics. M.A.K. wrote the sections on current knowledge of glacial isostasy and outstanding problems. D.A.W. wrote the section on rheology of the solid Earth. All authors commented on the final version of the article.

## Additional information

**Competing interests:** The authors declare no competing interests.

