## [Peer Review File · Nature Communications]

Reviewer #1 (Remarks to the Author):

Review of "Solid Earth processes and the evolution of the Antarctic Ice Sheet" by Pippa L. Whitehouse, Natalya Gomez, Matt A. King, and Douglas A. Wiens

The present paper reviews recent research into how the Antarctic Ice Sheet (AIS) interacts with the solid Earth as well as the bedrock. It first gives an overview over state-of-the-art quantification of glacial isostatic adjustment (GIA) and its impact on estimates of glacial mass balance. It then discusses variations and implications of the solid Earth rheology, both regionally based on seismic surveys and conceptually with respect to the choice of the material law. The next focus is the feedback between marine ice sheets and the solid Earth through water depth modulation at and around the grounding line, followed by an account of how the ice sheet shapes the topography mainly through erosion and a brief discussion how ice dynamics and volcanism interact with each other. The paper concludes with an account of where the authors identify the need for further research in this context.

Overall, I think, a review of the literature about these processes is timely and welcome and will both stimulate new research into the open questions raised by the authors and communicate the past findings to a larger audience. For example, the application of isostasy in numerical ice sheet models of Antarctica in the community of ice sheet modellers may well improve with respect to parameter choice or awareness of feedback mechanisms with this review at hand.

However, the outline and the structure of the paper seem a bit unfocused and exclusive in places, and there are a few things for me to raise for each of the sections that the authors should address before I fully recommend publication:

1) Title: Under this title, I would have expected to see a review of the effect of geothermal heat flux on the AIS in this paper, and possibly a more complete overview over erosion. These interactions are deliberately left out (L91-93), for which there are definitely good reasons (distribution of themes between this paper and the companion one, Colleoni et al., in review). However, the title should then reflect only those processes that are actually discussed here. I am not sure how to achieve this. One way could be to refer to processes controlling topography, but this would leave out the volcanism section.

2) "Motivation": While GIA is indeed the major process discussed in the paper, along with rheology, the motivation and specifically the opening should be more inclusive with respect to the other processes that are only brought up at the very end of the section. I suggest to go for why and how short and long-scale features in the bedrock topography affect ice flow, and then move on to how these evolve together with the ice sheet (with focus, obviously, on larger spatial scales, as stated by the authors).

3) "Current state of knowledge on GIA across Antarctica": Although not particularly a review article, and instead very much on the quantitative side, Martin-Español et al. (2016) have reviewed recent GIA solutions, with a certain overlap with this section. There are some solutions shown in Figure 1 of the present paper that are absent in Martin-Español et al. (2016). These might actually be very interesting ones because they partially account for 3D Earth structure and/or a coupled evolution like the solutions in Fig 1 f/g/h.

I suggest that the authors explain whether these presumably better new solutions do actually

advance the knowledge on GIA across Antarctica.

It is clear that the present paper does not aim for a similarly quantitative assessment of GIA solutions as Martin-Español et al. (2016), but some simple measure or even just a qualitative reasoning could help the reader understand how advance in the models actually improves their capabilities in simulating today's GIA signal.

4) "Rheology of the solid Earth beneath Antarctica": In my opinion, this section seems to be that part of the paper that connects most of the other sections (except probably the section on landscape evolution), which would possibly justify to move it to the more prominent place directly following the motivation.

The section would also benefit from some editing. For example, in the whole context of the Maxwell and Burger rheologies (L174-L197), no direct reference to Antarctica is made. There is also quite a lot of detail on the implementation of the discussed methods in places (for example L242-L244), and some redundant statements (like the reference to suggested lateral variations in L200, L204, L251-52). I suggest this section be tightened up, by removing redundancy and technical detail, and possibly by introducing an info box (I understand that these are allowed in Nature Communications review articles) about the conceptual aspects of rheology (Maxwell & Burger rheologies, dislocation and diffusion creep, ...).

5) "Feedbacks between GIA processes and ice dynamics": This section gives a very good account of the negative feedback that the deformational and gravitational response has on grounding line motion. With the review by de Boer et al. (2017) touching, among others, on this feedback, too, I suggest to focus this section more on Antarctica, e.g. by moving much of the first paragraph and Fig 4 to an info box, too.

Also, the section is exclusively about the sea level feedback, whereas its title refers to more general "GIA processes". I appreciate that the sea level feedback might be the most important one, and the most studied in recent years, but others have been proposed, too, like for example a feedback between ice sheet surface elevation and precipitation, or bedrock gradients and driving stress/flow velocities. These should at least be mentioned here. For example, Adhikari et al (2014) show how an ad hoc instantaneous change of bedrock, loosely related to GIA, would alter present day surface velocities and thus discharge. The elevation feedback is probably implemented in some way in the simulations from which the results in Fig 5 are taken and also in many other studies of longterm evolution of the AIS. For non-Antarctic ice sheets (and thus less relevant for this paper), the latter has also been studied in more detail by van den Berg (2008) and Konrad et al (2014).

6) "Feedbacks between Antarctic Ice Sheet change and long-term landscape evolution": Although this probably falls into the category of interaction at small spatial scale reviewed in the companion paper, it should be noted in this context that bedrock and ice stream flow interact significantly on much shorter timescales, too (e.g. Smith et al., 2007).

Consistency within the manuscript could be improved by canceling the "Antarctic Ice Sheet" from the section title as in previous and following section titles (or introduce AIS into these titles).

7) "Feedbacks between ice-sheet change and magmatic processes": As the references to interaction with the dynamic topography (Austermann et al., 2015) and thermal subsidence (Wilson & Luyendyk, 2009) seem a bit lost at the beginning of the previous section that mainly discusses erosion, deposition, and sedimentation, wouldn't it make sense to include these here in a broader (thermo-)dynamical scope?

The authors refer to studies which suggest enhanced volcanism at times of relative unloading and

respective feedbacks. The same would be true for volcanism in oceanic areas during times of sea level lows (Hasenclever et al., 2017), which would be partially and indirectly related to the evolution of the AIS. Could this have an impact on ice sheets, too?

8) "Outstanding Problems and Future Outlook": Again, this section discusses almost exclusively questions related to GIA. A more inclusive outlook could integrate content from the "landscape evolution" section here, such as parts of the paragraph starting at L439 ("would allow testing the hypothesis"), or from L472 ("development of such models will improve...").

Smaller comments

L7: "internal", this is misleading (internal ice-dynamical feedbacks or internal in the Earth system?), please clarify, also in L80.

L11-12: "position of the interface with the surrounding ocean"; if it at all fits into the word limit, I'd rather go for "position of the grounding line, the interface with the surrounding ocean, ...", or leave grounding line as stand-alone. Otherwise, in my experience, people will confuse this with the calving front.

L15-16: I absolutely agree with this statement, but it seems a bit detached here, and I would rather drop it or begin the abstract with this statement and take it from there.

L35: "gravity-field change"

L56: "(2002-2017)": Move to the end of the sentence?

L60: "dominant source of error in GRAVIMETRY-BASED estimates", or similar, as for example altimetry suffers more from the lack of constraint regarding the interaction with the atmosphere.

L87-88: As changes of the ice sheet bed due to erosion happen on much shorter time scales, too, please specify the focus to larger spatial scales (as in L91-93).

L109: This statement, for me, implies that the present article reviews reconstructions of past AIS extent in a similarly comprehensive way as for example rheology, which it does not. Instead, it is only the following paragraph and one in the "outstanding problems" section (L530-L543) that briefly summarize such reconstructions. Therefore, I suggest referring to more complete reviews of such reconstructions in the literature here, instead.

L115: "governed": restricted?

L148-149: Please explain how the sign affects mass balance estimates or drop "opposite sign".

L156 and L165: Could "region of Pine Island Glacier" be "Amundsen Sea embayment"? While "region" would technically include Thwaites et al., Pine Island Glacier is yet only of several relevant glaciers here.

L158-159: Please consider "... reflects THE WIDELY DOCUMENTED ice load change ..." or similar. The statement as it stands does not necessary imply that there are big changes going on in this area.

Please also include references.

L180-182: Please add a reference here.

L200-201: "... with greater heterogeneity than observed in areas of Northern Hemispheric glaciation.": Please add a reference here.

L249-50: Has this alternative approach been applied yet, and if so, by whom?

L271: Please drop "zones" here or explain why grounding lines are actually zones.

L272-74: The example of a "small rise in sea level" seems a bit random, why not stick to water depth?

L290: Ice shelf basal melt does not necessarily lead to retreat; "loss of buttressing/back stress following ice shelf basal melt" would be more accurate.

L299: The paragraph discusses mostly how ice shelves stabilize ice sheets, that ice rises enhance this support, and that water depth has an effect here. In my opinion, the introductory sentence, however, points more in the direction of processes involved in ice shelf break-up. So I would rather rephrase as "Water depth changes can also influence the stability of ice sheets via pinning of their attached ice shelves" or similar. Or actually introduce a discussion how ice rises inhibit damage propagation, if the authors are really after the processes involved in ice shelf stability.

L302: Please drop "particularly"; friction at the base occurs only to a non-negligible degree where local grounding is involved.

L316: I think the reference to Konrad et al. (2014) is unnecessary here (no ice shelves/grounding line involved, not Antarctica).

L351: "through the sea level feedback"; as the sea level feedback is not the only one (see above), this statement should be more inclusive, e.g. "through feedbacks".

L372: It is interesting that low viscosities, albeit unrealistic in East Antarctica, allow a faster re-advance (cf. Pollard et al., 2017, Fig. 10) as the enhanced sea level feedback in this case should, naively, act negatively on the advance, too. One would then assume that it is more a matter of pre-conditioning at the time when the readvance starts, and when low viscosities have had enabled much shallower oceans through faster rebound. It is worth to mention this here; otherwise the unacquainted reader might stumble upon this apparent contradiction. (This is similar to Chapter 5 in Konrad (2015), though the latter remains grey literature and thus does not need to be acknowledged here.)

L381-82: "... into the GIA corrections that should be applied to modern measurements of these systems." This is quite abstract; I suggest to either explicitly explain what "modern measurements of these systems" are, or to simply end this sentence with "into the GIA signal", as the contamination of mass balance estimates has been discussed in a previous section.

L382-384: This is already discussed at several instances in the paper, that in my opinion are better

suited for this statement (L158, L538, and related to this, L509), so it could be dropped here, or the wording could be moved to one of these instances so that the indeed important message is strengthened.

Figure 5: Switch positions of panels B and C?

In panel A, are ELRA and coupled models treated equally regarding the iteration of the initial topography, which can have significant impact on water depth and consequently ice sheet growth/decline? This, albeit quite a technical issue, would be an important piece of information for an ice sheet modeller who considers employing a coupled model instead of their ELRA implementation.

L411-412: "... alter the underlying topography on a more permanent basis", please rephrase.

L427: "By using..."

L431: "the West Antarctic Ice Sheet could have formed much earlier than previously though"; please specify the timing here.

L449: "... or by causing the system to pass some internal threshold.": please explain.

L479: The whole section on "magmatic processes" is relatively speculative, as the authors themselves acknowledge in the first and final sentences. In this light, I would like to see the characterization as "potentially the most dramatic" toned down a bit, but recognize that this is arguable.

L519-529: This paragraph would benefit from spelling out in more detail the lack of constraints and avenues for improvement, as it remains mostly vague and leaves a lot to the reader's expertise.

L519: It is not clear from the section on varying lateral Earth structure why "East Antarctica and all offshore regions" would stand out here, compared to West Antarctica. Please specify.

L520: "Beyond the coarse Earth structure", please specify.

L529: "but could be more complex in the shallow mantle and asthenosphere" Please explain.

L530: I'd prefer "extremely" or similar over "too", as "too" would require the specification of some criterion that cannot be met with the current amount of constraints. Or name this criterion and retain "too"?

L548: In this context, one could also mention the quantification of deformation using InSAR (e.g. Auriac et al, 2013).

Kind regards
Hannes Konrad

References

Adhikari, S., Ivins, E. R., Larour, E., Seroussi, H., Morlighem, M., and Nowicki, S.: Future Antarctic bed topography and its implications for ice sheet dynamics, *Solid Earth*, 5, 569-584, <https://doi.org/10.5194/se-5-569-2014>, 2014.

Auriac, A., K. H. Spaans, F. Sigmundsson, A. Hooper, P. Schmidt, and B. Lund (2013), Iceland rising: Solid Earth response to ice retreat inferred from satellite radar interferometry and viscoelastic modeling, *J. Geophys. Res. Solid Earth*, 118, 1331–1344, doi: 10.1002/jgrb.50082.

Austermann, J. et al. The impact of dynamic topography change on Antarctic ice sheet stability during the mid-Pliocene warm period. *Geology* 43, 927-930, doi:10.1130/G36988.1 (2015).

Colleoni, F. et al. Beneath Antarctica the ice-bed-ocean system: processes across the time scales. *Nat Commun* (in review).

de Boer, B., Stocchi, P., Whitehouse, P. L., and van de Wal, R.S.W. (2017), Current state and future perspectives on coupled ice-sheet - sea-level modelling, *Quat. Sci. Rev.* 169, 13-28, doi: 10.1016/j.quascirev.2017.05.013.

Hasenclever, Knorr, Rüpke, Köhler, Morgan, Garofalo, Barker, Lohmann & Hall (2017): Sea level fall during glaciation stabilized atmospheric CO₂ by enhanced volcanic degassing, *Nature Communications* 8, 15867, doi: 10.1038/ncomms15867.

Konrad, H. (2015): Sea-level and solid-Earth feedbacks on ice-sheet dynamics, PhD dissertation, Department of Earth Sciences, Free University of Berlin, http://www.diss.fu-berlin.de/diss/receive/FUDISS_thesis_00000099901.

Konrad, H., M. Thoma, I. Sasgen, V. Klemann, D. Barbi, K. Grosfeld, and Z. Martinec. The deformational response of a viscoelastic solid earth model coupled to a thermomechanical ice sheet model. *Surv. Geophys.*, 35(6):1441-1458, 2014. doi: 10.1007/s10712-013-9257-8.

Le Meur and Huybrechts (1996): A comparison of different ways of dealing with isostasy: examples from modelling the Antarctic ice sheet during the last glacial cycle. *Annals of Glaciology*, vol.23, pp.309-317.

Martin-Español, A., M. A. King, A. Zammit-Mangion, S. B. Andrews, P. Moore, and J. L. Bamber (2016), An assessment of forward and inverse GIA solutions for Antarctica, *J. Geophys. Res. Solid Earth*, 121, 6947-6965, doi: 10.1002/2016JB013154.

J. Oerlemans. Model experiments on the 100,000-yr glacial cycle. *Nature*, 287 (5781):430-432, 1980. doi: 10.1038/287430a0.

Smith, A., Murray, T. Nicholls, K., Makinson, K., Adalgeirsdóttir, G., Behar, A., and Vaughan, D. (2007). Rapid erosion, drumlin formation, and changing hydrology beneath an Antarctic ice stream. *Geology*. 35. 127-130. 10.1130/G23036A.1.

J. van den Berg, R.S.W. van de Wal, G.A. Milne, and J. Oerlemans. Effect of isostasy on dynamical ice sheet modeling: A case study for Eurasia. *J. Geophys. Res.*, 113:B05412, 2008. doi: 10.1029/2007JB004994.

Wilson, D. S. & Luyendyk, B. P. West Antarctic paleotopography estimated at the Eocene-Oligocene climate transition. *Geophys Res Lett* 36, L16302, doi:10.1029/2009gl039297 (2009).

Reviewer #2 (Remarks to the Author):

This is an interesting review paper, which really gives a great overview on feedbacks between the ice sheet and the solid Earth. The paper is already very well written and almost ready for publication. I have few minor comments:

Line 5: "significant contribution", add a number instead.

Line 39: perhaps you need to shortly describe what is "weak" Earth rheology.

Line 113-114:

How about GPS uplift rates? Do GIA models use GPS derived uplift rates as constraints?

Line 114

I think a map showing location of geological and geomorphological data (and GPS sites) uses as constraints could be very useful here. Also a map with area names and location of glaciers mentioned in the text will be useful.

Line 153: Add a reference to "Shepard et al, 2018, nature" (The IMBIE paper).

Line 343-344: "...leading to substantial (i.e., metres to tens of metres of) viscous uplift occurring on centennial or even decadal timescales). I am wondering, what is the elastic uplift here? Several meters? does it have implications for ice sheet evolution? Has anyone looked at that?

Line 415-416: perhaps you should mention that delta progradation driven by high freshwater runoff from the Antarctic Ice Sheet. For Greenland this is a significant effect even on relative short timescale (decades, see Bendixen et al, 2017, nature: doi:10.1038/nature23873)

Line 548: It will be nice to see where the new absolute gravity sites are located. Again, add a map it will really help getting an overview.

Best regards

Shfaqat Abbas Khan

Reviewer #3 (Remarks to the Author):

The report is excellent as far as it goes. However, the authors neglect the important area of a missing discussion of volcanism beneath the West Antarctic Ice Sheet (WAIS). Aeromagnetic surveys

combined with coincident radar ice sounding provides evidence for this which the authors apparently realize. However they refer to a manuscript (in review) which I consider inadequate. They note that deglaciation accelerates volcanism as in Iceland. I pointed this out in a paper I presented the Polar 2018 meeting in Davos a few weeks ago and at the International Glaciological Society meeting in Boulder last year. There is a great deal of evidence for subglacial volcanic rocks beneath the WAIS; some are active.

I recommend publication only after consideration of the material referenced as "in review".

Response to Editor's and Reviewers' comments

We thank the reviewers and the Editor for their insightful comments, and include our responses below in red. Note that we refer to line numbers in the previous version of our manuscript using the phrase "(original) line X".

Response to comments from the Editor

As you will see, on the whole the reviewers are very positive about the review and feel it provides a useful synthesis of the published literature. A number of points have, however, been raised, with the majority coming from Reviewer #1. We feel this is a particularly accurate assessment of the current status of the manuscript and would like you to follow the guidance given (in addition to the editorial suggestions below) when revising the Review.

We most particularly agree with Reviewer #1 in that, at present, the review could benefit from restructuring and refocusing in certain areas. There is currently a rather large focus on GIA/GIA modelling, with other aspects (volcanism, geotherm flux, erosion etc) taking more of a back seat. This is particularly problematic given the recent publication of <https://www.earth-surf-dynam.net/6/401/2018/>. While a very different article in terms of tone, there is considerable overlap in places (e.g. figure 4). We feel that refocusing the review in the broader context of the solid Earth system will help address this current imbalance. We provide some suggestions on how you might consider approaching this below. These are just suggestions, we fully acknowledge that you and your co-authors will be best placed to determine the best approach to take.

We thank the Editor for their constructive comments which have prompted us to revise the structure of the article. We have reduced the focus on glacial isostatic adjustment (GIA) modelling to ensure distinction with the earlier article. For information: Fig. 4 (now Fig. 5) depicts many more processes than the figure in the Earth Surface Dynamics article mentioned above, these two figures were drafted completely independently.

We recommend providing a shorter introduction/motivation that highlights the importance of understanding Antarctica/Solid Earth interactions, hints at current knowledge gaps (that will emerge in full during the rest of the review), followed by a brief outline of the intent of the review (i.e. assess current understanding and uncertainties).

The Introduction has been significantly shortened and refocused. Some material from deleted paragraphs of the Introduction has been incorporated into the section on "Current state of knowledge...", and some material from the "Current state of knowledge..." section has been incorporated into the section on solid Earth rheology.

As suggested by Reviewer #1, we feel it would then be beneficial to present the

individual solid Earth components (with the aid of a schematic) and the current understanding and remaining uncertainties of each aspect.

This has been addressed via the introduction of a new Fig. 1 and a rewrite of the beginning of the Introduction. We now describe a wider range of ice-solid Earth interactions in the opening paragraph, including basal heat flux and subglacial volcanism, but then make it clear that the primary focus of the article is on processes that result in long-wavelength reshaping of the bed of the Antarctic Ice Sheet, and the impact of these changes on ice sheet evolution.

GIA modelling could subsequently be woven into the narrative in the context of the degree to which improvements in understanding/remaining uncertainties regarding the solid Earth components/interactions have impacted/are impacting GIA modelling and our broader understanding of Antarctic Ice Sheet dynamics (drawing together components of the 'current state of knowledge...' and 'feedbacks between...' sections). At present we feel the Review lacks a comprehensive synthesis/reinterpretation aspect and approaching the GIA modelling component in this way (evaluating the respective importance of different solid Earth components) will go some way to providing this.

In the revised manuscript we focus on feedbacks between ice sheet evolution, glacial isostasy, and glacial erosion. We retain the broad structure of the previous version of the manuscript, but the focus on GIA has been reduced, and we seek to discuss the constituent processes rather than labelling something generically as 'GIA'.

By combining a detailed discussion of spatially variable Earth rheology with a quantitative analysis of the impact of Earth deformation and sea-level change on ice dynamics this article represents a re-interpretation of the current understanding of ice-Earth feedbacks across Antarctica. We feel that it is the most thorough synthesis to date of these two cutting-edge areas of contemporary research.

This will enable you, in the closing section of the review, to identify/prioritise areas for future research, possibly with a synthesis figure that demonstrates the relative importance of each aspect for different regions of Antarctica.

A figure has been included near the end of the manuscript that documents the location of key data sets around Antarctica. Location-specific priorities for future research are described in the adjacent text, but are not specifically labelled on the figure to avoid clutter.

Finally, you will note in Reviewer #1's comments, they propose the use of boxes on more than one occasion. While boxes are indeed possible in Nature Communications reviews, they should be used to provide additional information to the main narrative i.e. topics parallel to the main text that could provide additional helpful context. Further formatting requirements are explained at <http://www.nature.com/ncomms/submit/content-types>

Some text from the section on "Earth structure and rheology..." has been moved to an information box.

Reviewer #1 (Remarks to the Author):

Review of "Solid Earth processes and the evolution of the Antarctic Ice Sheet" by Pippa L. Whitehouse, Natalya Gomez, Matt A. King, and Douglas A. Wiens

The present paper reviews recent research into how the Antarctic Ice Sheet (AIS) interacts with the solid Earth as well as the bedrock. It first gives an overview over state-of-the-art quantification of glacial isostatic adjustment (GIA) and its impact on estimates of glacial mass balance. It then discusses variations and implications of the solid Earth rheology, both regionally based on seismic surveys and conceptually with respect to the choice of the material law. The next focus is the feedback between marine ice sheets and the solid Earth through water depth modulation at and around the grounding line, followed by an account of how the ice sheet shapes the topography mainly through erosion and a brief discussion how ice dynamics and volcanism interact with each other. The paper concludes with an account of where the authors identify the need for further research in this context.

Overall, I think, a review of the literature about these processes is timely and welcome and will both stimulate new research into the open questions raised by the authors and communicate the past findings to a larger audience. For example, the application of isostasy in numerical ice sheet models of Antarctica in the community of ice sheet modellers may well improve with respect to parameter choice or awareness of feedback mechanisms with this review at hand.

However, the outline and the structure of the paper seem a bit unfocused and exclusive in places, and there are a few things for me to raise for each of the sections that the authors should address before I fully recommend publication:

1) Title: Under this title, I would have expected to see a review of the effect of geothermal heat flux on the AIS in this paper, and possibly a more complete overview over erosion. These interactions are deliberately left out (L91-93), for which there are definitely good reasons (distribution of themes between this paper and the companion one, Colleoni et al., in review). However, the title should then reflect only those processes that are actually discussed here. I am not sure how to achieve this. One way could be to refer to processes controlling topography, but this would leave out the volcanism section.

The structure of the article has been revised to provide more focus, and this is reflected in the new title. We have removed the standalone section on magmatic processes and relevant text is incorporated elsewhere, e.g. the impact of subglacial volcanism and geothermal heat flux on ice dynamics are introduced briefly in the "Introduction" section. The majority of the article now focuses on processes that alter the bed topography of the ice sheet, and resulting feedbacks on ice dynamics.

2) "Motivation": While GIA is indeed the major process discussed in the paper, along with rheology, the motivation and specifically the opening should be more inclusive with respect to the other processes that are only brought up at the very end of the section. I suggest to go for why and how short and long-scale features in the bedrock topography affect ice flow, and then move on to how these evolve together with the ice sheet (with focus, obviously, on larger spatial scales, as stated by the authors).

We have followed the reviewer's suggestion and now begin the Introduction with a broader consideration of all the processes that constitute feedbacks between the solid Earth and the Antarctic Ice Sheet. We then justify our decision to focus on glacial isostasy and glacial erosion by noting that the long-wavelength topographic changes brought about by these processes will have the most significant impact on long-term ice-sheet evolution.

3) "Current state of knowledge on GIA across Antarctica": Although not particularly a review article, and instead very much on the quantitative side, Martin-Español et al. (2016) have reviewed recent GIA solutions, with a certain overlap with this section. There are some solutions shown in Figure 1 of the present paper that are absent in Martin-Español et al. (2016). These might actually be very interesting ones because they partially account for 3D Earth structure and/or a coupled evolution like the solutions in Fig 1 f/g/h. I suggest that the authors explain whether these presumably better new solutions do actually advance the knowledge on GIA across Antarctica. It is clear that the present paper does not aim for a similarly quantitative assessment of GIA solutions as Martin-Español et al. (2016), but some simple measure or even just a qualitative reasoning could help the reader understand how advance in the models actually improves their capabilities in simulating today's GIA signal.

A table has been added that summarizes the advantages and limitations of the different approaches used to generate the GIA solutions shown in Fig. 1 (now Fig. 2).

4) "Rheology of the solid Earth beneath Antarctica": In my opinion, this section seems to be that part of the paper that connects most of the other sections (except probably the section on landscape evolution), which would possibly justify to move it to the more prominent place directly following the motivation.

We feel that it is important to first summarize the current state of knowledge on ice-sheet change and glacial isostasy before focusing on solid Earth rheology. The opening few sections has been significantly edited, and we welcome feedback on whether the new structure is an improvement.

The section would also benefit from some editing. For example, in the whole context of the Maxwell and Burger rheologies (L174-L197), no direct reference to Antarctica is made. There is also quite a lot of detail on the implementation of the discussed methods in places (for example L242-L244), and some redundant statements (like the reference to suggested lateral variations in L200, L204, L251-52). I suggest this section be tightened up, by removing redundancy and technical detail, and possibly by introducing an info box (I understand that these are allowed in Nature Communications review articles) about the conceptual aspects of rheology (Maxwell & Burger rheologies, dislocation and diffusion creep, ...).

Text from (original) lines 174-197 has been moved to an information box. This text describes the different rheological models that can be used to study the response of the solid Earth to surface load changes.

With regard to the text on the implementation of discussed methods: This has been revised to remove some of the technical detail.

With regard to the text on lateral variations: the statements on (original) lines 200 and 204 refer to variations in seismic velocities, material properties, and mantle viscosity, which are not the same thing. The text has been slightly edited to clarify this. The text on (original) lines 251-252 has been deleted.

5) "Feedbacks between GIA processes and ice dynamics": This section gives a very good account of the negative feedback that the deformational and gravitational response has on grounding line motion. With the review by de Boer et al. (2017) touching, among others, on this feedback, too, I suggest to focus this section more on Antarctica, e.g. by moving much of the first paragraph and Fig 4 to an info box, too.

We have decided not to move this material to an information box as it is central to the overall focus of the article. Text has been added to link this material more directly to the situation in Antarctica.

Also, the section is exclusively about the sea level feedback, whereas its title refers to more general "GIA processes". I appreciate that the sea level feedback might be the most important one, and the most studied in recent years, but others have been proposed, too, like for example a feedback between ice sheet surface elevation and precipitation, or bedrock gradients and driving stress/flow velocities. These should at least be mentioned here. For example, Adhikari et al (2014) show how an ad hoc instantaneous change of bedrock, loosely related to GIA, would alter present day surface velocities and thus discharge. The elevation feedback is probably implemented in some way in the simulations from which the results in Fig 5 are taken and also in many other studies of long-term evolution of the AIS. For non-Antarctic ice sheets (and thus less relevant for this paper), the latter has also been studied in more detail by van den Berg (2008) and Konrad et al (2014).

We already reference the article by Adhikari et al. (2014), and have edited the text in the accompanying sentence to clarify that we use this reference as an example of the effect of a change in bedrock configuration on ice dynamics.

Feedbacks between isostatically-driven surface elevation change and surface mass balance are now mentioned in the same paragraph: "An important process that is also accounted for in these coupled models is the feedback between isostatically-driven ice surface elevation change and surface mass", and appropriate references are included.

6) "Feedbacks between Antarctic Ice Sheet change and long-term landscape evolution": Although this probably falls into the category of interaction at small spatial scale reviewed in the companion paper, it should be noted in this context that bedrock and ice stream flow interact significantly on much shorter timescales, too (e.g. Smith et al., 2007).

Text near the end of the Introduction is edited to reflect the fact that such small scale processes are discussed elsewhere: "...but smaller-scale subglacial controls, such as the material properties of the bed and variations in subglacial hydrology and geomorphology are discussed elsewhere. "

Consistency within the manuscript could be improved by cancelling the "Antarctic Ice Sheet" from the section title as in previous and following section titles (or introduce AIS into these titles).

The text "Antarctic Ice Sheet change" has been replaced with "ice sheet change" in this section title.

7) "Feedbacks between ice-sheet change and magmatic processes": As the references to interaction with the dynamic topography (Austermann et al., 2015) and thermal subsidence (Wilson & Luyendyk, 2009) seem a bit lost at the beginning of the previous section that mainly discusses erosion, deposition, and sedimentation, wouldn't it make sense to include these here in a broader (thermo-)dynamical scope?

We prefer to retain all references to long-timescale processes in the section on long-term landscape evolution. The section on magmatic processes has now been deleted, with relevant material included elsewhere.

The authors refer to studies which suggest enhanced volcanism at times of relative unloading and respective feedbacks. The same would be true for volcanism in oceanic areas during times of sea level lows (Hasenclever et al., 2017), which would be partially and indirectly related to the evolution of the AIS. Could this have an impact on ice sheets, too?

Feedbacks between global sea-level change and volcanism would indirectly influence ice sheet evolution due to the link between volcanism and atmospheric CO₂. However, we chose not to discuss such feedbacks in this article because the link to ice sheet dynamics is fairly indirect, and this issue has also recently been covered elsewhere (Whitehouse, Earth Surface Dynamics, 2018 – although we were unaware of the article by Hasenclever et al., so will chase this up, thanks!).

8) "Outstanding Problems and Future Outlook": Again, this section discusses almost exclusively questions related to GIA. A more inclusive outlook could integrate content from the "landscape evolution" section here, such as parts of the paragraph starting at

L439 ("would allow testing the hypothesis"), or from L472 ("development of such models will improve...").

We now frame this section as providing a summary of priorities for future research that will improve our understanding feedbacks between solid Earth and ice sheet dynamics and our ability to quantify present-day ice sheet change across Antarctica. We acknowledge that many of the points discussed necessarily relate to improving constraints on the GIA signal. A broader summary of the whole article is included in the final section of the manuscript.

Smaller comments

L7: "internal", this is misleading (internal ice-dynamical feedbacks or internal in the Earth system?), please clarify, also in L80.

Text edited – the term "internal" has been deleted on (original) line 7 and replaced on (original) line 80.

L11-12: "position of the interface with the surrounding ocean"; if it at all fits into the word limit, I'd rather go for "position of the grounding line, the interface with the surrounding ocean, ...", or leave grounding line as stand-alone. Otherwise, in my experience, people will confuse this with the calving front.

Text edited to specifically refer to the grounding line.

L15-16: I absolutely agree with this statement, but it seems a bit detached here, and I would rather drop it or begin the abstract with this statement and take it from there.

Sentence deleted.

L35: "gravity-field change"

Text has been deleted.

L56: "(2002-2017)": Move to the end of the sentence?

Dates moved to the end of the sentence. Note that the paragraph containing this text has been moved.

L60: "dominant source of error in GRAVIMETRY-BASED estimates", or similar, as for example altimetry suffers more from the lack of constraint regarding the interaction with the atmosphere.

Text edited. Note that the paragraph containing this text has been moved.

L87-88: As changes of the ice sheet bed due to erosion happen on much shorter time scales, too, please specify the focus to larger spatial scales (as in L91-93).

We prefer not to specify the spatial scale of these changes, but instead retain the text that lists the processes responsible for change.

L109: This statement, for me, implies that the present article reviews reconstructions of past AIS extent in a similarly comprehensive way as for example rheology, which it does not. Instead, it is only the following paragraph and one in the "outstanding problems" section (L530-L543) that briefly summarize such reconstructions. Therefore, I suggest referring to more complete reviews of such reconstructions in the literature here, instead.

This sentence has been deleted, although we have checked that we include sufficient references later in the section to provide a comprehensive review of the current state of knowledge on reconstructing past Antarctic Ice Sheet change.

L115: "governed": restricted?

Text edited.

L148-149: Please explain how the sign affects mass balance estimates or drop "opposite sign".

Relevant text deleted.

L156 and L165: Could "region of Pine Island Glacier" be "Amundsen Sea embayment"? While "region" would technically include Thwaites et al., Pine Island Glacier is yet only of several relevant glaciers here.

Text edited to refer to the Amundsen Sea embayment.

L158-159: Please consider "... reflects THE WIDELY DOCUMENTED ice load change ..." or similar. The statement as it stands does not necessarily imply that there are big changes going on in this area. Please also include references.

Text edited to refer to "significant decadal-to-centennial ice load change", and a reference to the now-published article by Barletta et al. (2018) has been added.

L180-182: Please add a reference here.

We now include a reference to van der Wal et al., 2013 ("Glacial isostatic adjustment model with composite 3-D Earth rheology for Fennoscandia", *Geophysical Journal International*, 194, 61–77). Note that the accompanying text has been moved to the information box.

L200-201: "... with greater heterogeneity than observed in areas of Northern Hemispheric glaciation.": Please add a reference here.

We now include a reference to Schaeffer and Lebedev, 2013 ("Global shear speed structure of the upper mantle and transition zone", *Geophysical Journal International*, 194, 417-449) to highlight the low spatial variability in seismic upper mantle properties across continental regions of the northern hemisphere.

L249-50: Has this alternative approach been applied yet, and if so, by whom?

This approach has not yet been applied. Text edited to read "An alternative approach would involve..."

L271: Please drop "zones" here or explain why grounding lines are actually zones.

Text deleted.

L272-74: The example of a "small rise in sea level" seems a bit random, why not stick to water depth?

Text edited to refer to "water depth".

L290: Ice shelf basal melt does not necessarily lead to retreat; "loss of buttressing/back stress following ice shelf basal melt" would be more accurate.

Text edited to refer to "a reduction in ice shelf buttressing".

L299: The paragraph discusses mostly how ice shelves stabilize ice sheets, that ice rises enhance this support, and that water depth has an effect here. In my opinion, the introductory sentence, however, points more in the direction of processes involved in ice shelf break-up. So I would rather rephrase as "Water depth changes can also influence the stability of ice sheets via pinning of their attached ice shelves" or similar. Or actually introduce a discussion how ice rises inhibit damage propagation, if the authors are really after the processes involved in ice shelf stability.

Text edited to: "Water depth changes can also influence the degree to which ice shelves are able to stabilise the ice sheet".

L302: Please drop "particularly"; friction at the base occurs only to a non-negligible degree where local grounding is involved.

The term "particularly" is used to make it clear that the second half of the sentence just relates to friction at the base of the ice sheet. Text unchanged.

L316: I think the reference to Konrad et al. (2014) is unnecessary here (no ice shelves/grounding line involved, not Antarctica).

Reference to Konrad et al. (2014) has been removed from this sentence.

L351: "through the sea level feedback"; as the sea level feedback is not the only one (see above), this statement should be more inclusive, e.g. "through feedbacks".

Text deleted.

L372: It is interesting that low viscosities, albeit unrealistic in East Antarctica, allow a faster re-advance (cf. Pollard et al., 2017, Fig. 10) as the enhanced sea level feedback in this case should, naively, act negatively on the advance, too. One would then assume that it is more a matter of pre-conditioning at the time when the readvance starts, and when low viscosities have had enabled much shallower oceans through faster rebound. It is worth to mention this here; otherwise the unacquainted reader might stumble upon this apparent contradiction. (This is similar to Chapter 5 in Konrad (2015), though the latter remains grey literature and thus does not need to be acknowledged here.)

A sentence has been added to the end of the opening paragraph of this whole section on ice-Earth feedbacks, explaining that "Ongoing viscous uplift of the bed following a halt in ice sheet retreat can also initiate readvance of the grounding line in marine areas (e.g. Fig 6d, Pollard et al. 2017)."

L381-82: "... into the GIA corrections that should be applied to modern measurements of these systems." This is quite abstract; I suggest to either explicitly explain what "modern measurements of these systems" are, or to simply end this sentence with "into the GIA signal", as the contamination of mass balance estimates has been discussed in a previous section.

Text edited to refer to "contemporary geodetic observations".

L382-384: This is already discussed at several instances in the paper, that in my opinion are better suited for this statement (L158, L538, and related to this, L509), so it could be dropped here, or the wording could be moved to one of these instances so that the indeed important message is strengthened.

This text (in fact, the remainder of this paragraph) has been deleted as the material is covered later in the "Outstanding Problems and Future Outlook" section.

Figure 5: Switch positions of panels B and C?

Panel positions have been switched.

In panel A, are ELRA and coupled models treated equally regarding the iteration of the

initial topography, which can have significant impact on water depth and consequently ice sheet growth/decline? This, albeit quite a technical issue, would be an important piece of information for an ice sheet modeller who considers employing a coupled model instead of their ELRA implementation.

For the coupled and ELRA simulations shown in (renumbered) Fig. 6a the final topography is close to the modern observed topography, but the starting bedrock elevation is different in all three cases since the deformation over the course of the simulations is different. This also holds true for two different fully-coupled simulations with differing viscosity structure or lithospheric thickness. We agree that it is important to mention this technical point relating to initial bedrock elevation, and have added the following sentence: "Note that the bed topography at the start of each simulation shown in Fig. 6a will be different; this is to ensure that the final modelled topography is close to the modern observed topography in each case."

L411-412: "... alter the underlying topography on a more permanent basis", please rephrase.

Text edited to: "... several other processes also alter the underlying topography."

L427: "By using..."

Text edited.

L431: "the West Antarctic Ice Sheet could have formed much earlier than previously though"; please specify the timing here.

Text added to specify timing.

L449: "... or by causing the system to pass some internal threshold.": please explain.

Text has been added to further explain this point.

L479: The whole section on "magmatic processes" is relatively speculative, as the authors themselves acknowledge in the first and final sentences. In this light, I would like to see the characterization as "potentially the most dramatic" toned down a bit, but recognize that this is arguable.

Section has been deleted.

L519-529: This paragraph would benefit from spelling out in more detail the lack of constraints and avenues for improvement, as it remains mostly vague and leaves a lot to the reader's expertise.

The text in this paragraph has been tightened up.

L519: It is not clear from the section on varying lateral Earth structure why "East Antarctica and all offshore regions" would stand out here, compared to West Antarctica. Please specify.

Text added: "... due to significant uncertainty in Earth structure resulting from the absence of seismic stations."

L520: "Beyond the coarse Earth structure", please specify.

Text deleted.

L529: "but could be more complex in the shallow mantle and asthenosphere" Please explain.

Text has been added to clarify this point: "...could be more complex in the shallow mantle and asthenosphere due to ongoing isostatic relaxation or earthquake-related deformation".

L530: I'd prefer "extremely" or similar over "too", as "too" would require the specification of some criterion that cannot be met with the current amount of constraints. Or name this criterion and retain "too"?

Text edited to "extremely".

L548: In this context, one could also mention the quantification of deformation using InSAR (e.g. Auriac et al, 2013).

A reference to the Auriac et al. (2013) study has been added.

Kind regards
Hannes Konrad

References

Adhikari, S., Ivins, E. R., Larour, E., Seroussi, H., Morlighem, M., and Nowicki, S.: Future Antarctic bed topography and its implications for ice sheet dynamics, *Solid Earth*, 5, 569-584, <https://doi.org/10.5194/se-5-569-2014>, 2014.

Auriac, A., K. H. Spaans, F. Sigmundsson, A. Hooper, P. Schmidt, and B. Lund (2013),

Iceland rising: Solid Earth response to ice retreat inferred from satellite radar interferometry and viscoelastic modeling, *J. Geophys. Res. Solid Earth*, 118, 1331–1344, doi: 10.1002/jgrb.50082.

Austermann, J. et al. The impact of dynamic topography change on Antarctic ice sheet stability during the mid-Pliocene warm period. *Geology* 43, 927-930, doi:10.1130/G36988.1 (2015).

Colleoni, F. et al. Beneath Antarctica the ice-bed-ocean system: processes across the time scales. *Nat Commun* (in review).

de Boer, B., Stocchi, P., Whitehouse, P. L., and van de Wal, R.S.W. (2017), Current state and future perspectives on coupled ice-sheet - sea-level modelling, *Quat. Sci. Rev.* 169, 13-28, doi: 10.1016/j.quascirev.2017.05.013.

Hasenclever, Knorr, Rüpke, Köhler, Morgan, Garofalo, Barker, Lohmann & Hall (2017): Sea level fall during glaciation stabilized atmospheric CO₂ by enhanced volcanic degassing, *Nature Communications* 8, 15867, doi: 10.1038/ncomms15867.

Konrad, H. (2015): Sea-level and solid-Earth feedbacks on ice-sheet dynamics, PhD dissertation, Department of Earth Sciences, Free University of Berlin, http://www.diss.fu-berlin.de/diss/receive/FUDISS_thesis_000000099901.

Konrad, H., M. Thoma, I. Sasgen, V. Klemann, D. Barbi, K. Grosfeld, and Z. Martinec. The deformational response of a viscoelastic solid earth model coupled to a thermomechanical ice sheet model. *Surv. Geophys.*, 35(6):1441-1458, 2014. doi: 10.1007/s10712-013-9257-8.

Le Meur and Huybrechts (1996): A comparison of different ways of dealing with isostasy: examples from modelling the Antarctic ice sheet during the last glacial cycle. *Annals of Glaciology*, vol.23, pp.309-317.

Martin-Español, A., M. A. King, A. Zammit-Mangion, S. B. Andrews, P. Moore, and J. L. Bamber (2016), An assessment of forward and inverse GIA solutions for Antarctica, *J. Geophys. Res. Solid Earth*, 121, 6947-6965, doi: 10.1002/2016JB013154.

J. Oerlemans. Model experiments on the 100,000-yr glacial cycle. *Nature*, 287 (5781):430-432, 1980. doi: 10.1038/287430a0.

Smith, A., Murray, T. Nicholls, K., Makinson, K., Adalgeirsdóttir, G., Behar, A., and Vaughan, D. (2007). Rapid erosion, drumlin formation, and changing hydrology beneath an Antarctic ice stream. *Geology*. 35. 127-130. 10.1130/G23036A.1.

J. van den Berg, R.S.W. van de Wal, G.A. Milne, and J. Oerlemans. Effect of isostasy on dynamical ice sheet modeling: A case study for Eurasia. *J. Geophys. Res.*, 113:B05412, 2008. doi: 10.1029/2007JB004994.

Wilson, D. S. & Luyendyk, B. P. West Antarctic paleotopography estimated at the Eocene-Oligocene climate transition. *Geophys Res Lett* 36, L16302, doi:10.1029/2009gl039297 (2009).

Reviewer #2 (Remarks to the Author):

This is an interesting review paper, which really gives a great overview on feedbacks between the ice sheet and the solid Earth. The paper is already very well written and almost ready for publication. I have few minor comments:

Line 5: "significant contribution", add a number instead.

Text edited to quantify the potential sea-level contribution from Antarctica over the next few centuries.

Line 39: perhaps you need to shortly describe what is "weak" Earth rheology.

This whole paragraph has been significantly edited and the phrase "weak Earth rheology" is no longer used.

Line 113-114: How about GPS uplift rates? Do GIA models use GPS derived uplift rates as constraints?

Text edited to include reference to geodetic constraints in addition to geological and geomorphological constraints.

Line 114: I think a map showing location of geological and geomorphological data (and GPS sites) uses as constraints could be very useful here. Also a map with area names and location of glaciers mentioned in the text will be useful.

A figure (Fig. 7) showing the location of current GPS sites and data relating to past ice extent and sea level has been added. Locations mentioned in the text are labelled on this figure.

Line 153: Add a reference to "Shepard et al, 2018, nature" (The IMBIE paper).

A full reference to Shepherd et al. (2018) is now included.

Line 343-344: "...leading to substantial (i.e., metres to tens of metres of) viscous uplift occurring on centennial or even decadal timescales). I am wondering, what is the elastic uplift here? Several meters? does it have implications for ice sheet evolution? Has anyone looked at that?

Numerical models (including those used to produce the results shown in Fig. 6) typically predict the combined elastic and viscous response to surface load change, i.e. the viscoelastic response. We therefore now refer to "viscoelastic uplift" in this sentence and further discussion of the timing and magnitude of viscous vs. elastic uplift is included in the "Outstanding Problems and Future outlook" section of the article.

Line 415-416: perhaps you should mention that delta progradation driven by high freshwater runoff from the Antarctic Ice Sheet. For Greenland this is a significant effect even on relative short timescale (decades, see Bendixen et al, 2017, nature: doi:10.1038/nature23873)

I cannot find any studies that explore the development of fluvial deltas in Antarctica since the inception of the ice sheet. Freshwater runoff is a very limited component of the surface mass balance in Antarctica. Text not edited.

Line 548: It will be nice to see where the new absolute gravity site are located. Again, add a map it will really help getting an overview.

We already included a reference to an article from 2007 that reviews the status of absolute gravimetry measurements in Antarctica at the time. We are not aware of any more recent studies that consider how absolute gravity can be used to constrain the GIA signal in Antarctica; given the lack of up-to-date information on the status of this network it is not included in Fig. 7.

Best regards
Shfaqat Abbas Khan

Reviewer #3 (Remarks to the Author):

The report is excellent as far as it goes. However, the authors neglect the important area of a missing discussion of volcanism beneath the West Antarctic Ice Sheet (WAIS). Aeromagnetic surveys combined with coincident radar ice sounding provides evidence for this which the authors apparently realize. However they refer to a manuscript (in review) which I consider inadequate. They note that deglaciation accelerates volcanism as in Iceland. I pointed this out in a paper I presented the Polar 2018 meeting in Davos a few weeks ago and at the International Glaciological Society meeting in Boulder last

year. There is a great deal of evidence for subglacial volcanic rocks beneath the WAIS; some are active.

Unfortunately, in response to comments from the Editor and Reviewer 1, the standalone section on subglacial volcanism has now been deleted. We acknowledge the importance of this subject in the Introduction and include relevant references for those who would like to explore the subject further. Some of the text on subglacial volcanism from the original version of the manuscript has been moved to other sections, but we do not have space to include a specific section on this subject in this article.

I recommend publication only after consideration of the material referenced as "in review".

Only articles that have been published are included in the revised version of the text.

Reviewer #1 (Remarks to the Author):

Review of "Solid Earth change and the evolution 1 of the Antarctic Ice Sheet" by Pippa L. Whitehouse and coauthors (NCOMMS-17-31710B)

The authors have gone through a comprehensive revision of the manuscript. The new structure is much improved, with a clearer message on what is and what is not part of the review. I absolutely agree with the authors that "it is the most thorough synthesis to date of these two cutting-edge areas of contemporary research." The additional figures and the table will help the reader get insight into their specific contexts. As stated in my previous review report, I am sure that it will very effectively communicate open questions and past achievements to a larger audience and by that advance future research into the evolution of the Antarctic ice sheet and possibly beyond. In this sense, I gladly recommend publication of the manuscript in Nature Communications.

The points that I raise below are quite minor and might well depend on personal preferences.

Best regards
Hannes Konrad

L68-69: "by recent ice sheet-change (few millennia or less)", i.e. swap bracket and ice sheet change?

L234: "correcting for estimated geographical and depth variations in mantle composition", rephrase as "correcting for variations in mantle composition at estimated locations", or similar?

L255: "Crucially, such models typically include no lateral variation in rheological structure. However, ..." After "however", one would expect a statement that such lateral variation is indeed present in the Earth's interior. The following sentence(s) about dislocation creep and diffusion creep imply this, but it could be stated more clearly here.

L316 & L 317: "sea-level fall/rise", other instances of "sea level" in this context have been replaced by "water depth" during revision; I think this should be done here, too. For example, an ice rise's effectiveness in providing backstress to the ice sheet will also be enhanced by rebound of the ocean floor in the cavity.

L318: add "of the ice sheet" at the end of the sentence? The ice rise is confined by a grounding line, too, and they might be easily confused here.

L330-334: I acknowledge that technically my point about other feedbacks than water depth/grounding line has been resolved. However, I still feel that the full scope of interaction mechanisms is under-represented. The reference to "GIA-related sea level and solid Earth changes" might be interpreted by readers as a reference to the one feedback mechanism only (water depth/grounding line). Why not extend the final sentence of the paragraph in order to communicate that changing bedrock slopes imply changing driving stress/velocity field?

L456: "between 34 and 14 Ma THE ANTARCTIC ice sheet volume fluctuated"?

Response to Reviewer's comments

We thank Reviewer 1 for their comments on our revised article, and address each of their points below. Our responses are in blue.

L68-69: "by recent ice sheet-change (few millennia or less)", i.e. swap bracket and ice sheet change?

Text edited as suggested.

L234: "correcting for estimated geographical and depth variations in mantle composition", rephrase as "correcting for variations in mantle composition at estimated locations", or similar?

Text edited to "...correcting for (poorly known) spatial variations in mantle composition."

L255: "Crucially, such models typically include no lateral variation in rheological structure. However, ..." After "however", one would expect a statement that such lateral variation is indeed present in the Earth's interior. The following sentence(s) about dislocation creep and diffusion creep imply this, but it could be stated more clearly here.

A sentence has been added to address this point: "However, differences in the response of the Earth to surface loading around the world suggest regional variations in rheological properties."

L316 & L 317: "sea-level fall/rise", other instances of "sea level" in this context have been replaced by "water depth" during revision; I think this should be done here, too. For example, an ice rise's effectiveness in providing backstress to the ice sheet will also be enhanced by rebound of the ocean floor in the cavity.

Text has been revised to refer to 'water depth' rather than 'sea level': "A local decrease in water depth can enhance grounding of the ice shelf at ice rises, stabilising the ice sheet, while an increase in water depth can lead to ungrounding at the ice rise..."

L318: add "of the ice sheet" at the end of the sentence? The ice rise is confined by a grounding line, too, and they might be easily confused here.

Text edited as suggested.

L330-334: I acknowledge that technically my point about other feedbacks than water depth/grounding line has been resolved. However, I still feel that the full scope of interaction mechanisms is under-represented. The reference to "GIA-related sea level and solid Earth changes" might be interpreted by readers as a reference to the one feedback mechanism only (water depth/grounding line). Why not extend the final sentence of the paragraph in order to communicate that changing bedrock slopes imply changing driving stress/velocity field?

Text has been edited to read: "It has been shown that GIA-related sea-level and solid Earth changes, including changes to the slope of the underlying bed, alter the stress field of the ice sheet in a way that acts to dampen and slow..."

L456: "between 34 and 14 Ma THE ANTARCTIC ice sheet volume fluctuated"?

Text edited to: "...between 34 and 14 Ma the volume of the AIS fluctuated significantly..."